# Graph Diffusion Policy Optimization

**Yijing Liu**[*1], **Chao Du**[*†2], **Tianyu Pang**[2], **Chongxuan Li**[3], **Min Lin**[2], **Wei Chen**[†1]

[1]State Key Lab of CAD&CG, Zhejiang University
[2]Sea AI Lab, Singapore
[3]Renmin University of China
{liuyj86,chenvis}@zju.edu.cn;
{duchao,tianyupang,linmin}@sea.com; chongxuanli@ruc.edu.cn

## Abstract

Recent research has made significant progress in optimizing diffusion models for downstream objectives, which is an important pursuit in fields such as graph generation for drug design. However, directly applying these models to graph presents challenges, resulting in suboptimal performance. This paper introduces *graph diffusion policy optimization* (GDPO), a novel approach to optimize graph diffusion models for arbitrary (e.g., non-differentiable) objectives using reinforcement learning. GDPO is based on an *eager policy gradient* tailored for graph diffusion models, developed through meticulous analysis and promising improved performance. Experimental results show that GDPO achieves state-of-the-art performance in various graph generation tasks with complex and diverse objectives. Code is available at https://github.com/sail-sg/GDPO.

## 1 Introduction

Graph generation, a key facet of graph learning, has applications in a variety of domains, including drug and material design [54], code completion [8], social network analysis [20], and neural architecture search [64]. Numerous studies have shown significant progress in graph generation with deep generative models [34, 62, 69, 21]. one of The most notable advances in the field is the introduction of graph diffusion probabilistic models (DPMs) [61, 31]. These methods can learn the underlying distribution from graph data samples and produce high-quality novel graph structures.

In many use cases of graph generation, the primary focus is on achieving specific objectives, such as high drug efficacy [60] or creating novel graphs with special discrete properties [22]. These objectives are often expressed as specific reward signals, such as binding affinity [10] and synthetic accessibility [7], rather than a set of training graph samples. Therefore, a more pertinent goal in such scenarios is to train graph generative models to meet these predefined objectives directly, rather than learning to match a distribution over training data [72].

A major challenge in this context is that most signals are non-differentiable w.r.t. graph representations, making it difficult to apply many optimization algorithms. To address this, methods based on property predictors [29, 37] learn parametric models to predict the reward signals, providing gradient guidance for graph generation. However, since reward signals can be highly complex (e.g., results from physical simulations), these predictors often struggle to provide accurate guidance [44]. An alternative direction is to learn graph generative models as policies through reinforcement learning (RL) [72], which enables the integration of exact reward signals into the optimization. However, existing work primarily explores earlier graph generative models and has yet to leverage the superior performance of graph DPMs [9, 68]. On the other hand, several pioneer works have seen significant

---

[*]Equal contribution. Work done during Yijing Liu's internship at Sea AI Lab.
[†]Correspondence to Wei Chen and Chao Du.

38th Conference on Neural Information Processing Systems (NeurIPS 2024).

progress in optimizing continuous-variable (e.g., images) DPMs for downstream objectives [6, 16]. The central idea is to formulate the sampling process as a policy, with the objective serving as a reward, and then learn the model using policy gradient methods. However, when these approaches are directly extended to (discrete-variable) graph DPMs, we empirically observe a substantial failure, which we will illustrate and discuss in Sec. 4.

To close this gap, we present *graph diffusion policy optimization* (GDPO), a policy gradient method designed to optimize graph DPMs for arbitrary reward signals. Using an RL formulation similar to that introduced by Black et al. [6] and Fan et al. [16] for continuous-variable DPMs, we first adapt the discrete diffusion process of graph DPMs to a Markov decision process (MDP) and formulate the learning problem as policy optimization. Then, to address the observed empirical failure, we introduce a slight modification to the standard policy gradient method REINFORCE [58], dubbed the *eager policy gradient* and specifically tailored for graph DPMs. Experimental evaluation shows that GDPO proves effective across various scenarios and achieves high sample efficiency. Remarkably, our method achieves a $\mathbf{41.64\%}$ to $\mathbf{81.97\%}$ average reduction in generation-test distance and a $1.03\%$ to $\mathbf{19.31\%}$ improvement in the rate of generating effective drugs, while only querying a small number of samples (1/25 of the training samples).

## 2 Related Works

**Graph Generative Models.** Early work in graph generation employs nonparametric random graph models [15, 26]. To learn complex distributions from graph-structured data, recent research has shifted towards leveraging deep generative models. This includes approaches based on auto-regressive generative models [69, 39], variational autoencoders (VAEs) [34, 41, 23], generative adversarial networks (GANs) [62, 9, 43], and normalizing flows [53, 40, 42].

Recently, diffusion probabilistic models (DPMs) [25, 56] have significantly advanced graph generation [70]. Models like EDP-GNN [46] GDSS [31] and DruM [30] construct graph DPMs using continuous diffusion processes [57]. While effective, the use of continuous representations and Gaussian noise can hurt the sparsity of generated graphs. DiGress [61] employs categorical distributions as the Markov transitions in discrete diffusion [2], performing well on complex graph generation tasks. While these works focus on learning graph DPMs from a given dataset, our primary focus in this paper is on learning from arbitrary reward signals.

**Controllable Generation for Graphs.** Recent progress in controllable generation has also enabled graph generation to achieve specific objectives or properties. Previous work leverages mature conditional generation techniques from GANs and VAEs [66, 52, 36, 28, 14]. This paradigm has been extended with the introduction of guidance-based conditional generation in DPMs [12]. DiGress [61] and GDSS [31] provide solutions that sample desired graphs with guidance from additional property predictors. MOOD [37] improves these methods by incorporating out-of-distribution control. However, as predicting the properties (e.g., drug efficacy) can be extremely difficult [33, 44], the predictors often struggle to provide accurate guidance. Our work directly performs property optimization on graph DPMs, thus bypassing this challenge.

**Graph Generation using RL.** RL techniques find wide application in graph generation to meet downstream objectives. REINVENT [47] and GCPN [68] are representative works, which define graph environments and optimize policy networks with policy gradient methods [59]. For data-free generation modelling, MolDQN [71] replaces the data-related environment with a human-defined graph environmentand and utilizes Q-Learning [24] for policy optimi zation. To generate more realistic molecules, DGAPN [63] and FREED [67] investigate the fragment-based chemical environment, which reduce the search space significantly. Despite the great successes, existing methods exhibit high time complexity and limited policy model capabilities. Our work, based on graph DPMs with enhanced policy optimization, achieves new state-of-the-art performance.

**Aligning DPMs.** Several works focus on optimizing generative models to align with human preferences [45, 3]. DPOK [16] and DDPO [6] are representative works that align text-to-image DPMs with black-box reward signals. They formulate the denoising process of DPMs as an MDP and optimize the model using policy gradient methods. For differentiable rewards, such as human preference models [35], AlignProp [50] and DRaFT [11] propose effective approaches to optimize DPMs with direct backpropagation, providing a more accurate gradient estimation than DDPO and DPOK. However,

these works are conducted on images. To the best of our knowledge, our work is the first effective method for aligning graph DPMs, filling a notable gap in the literature.

# 3 Preliminaries

In this section, we briefly introduce the background of graph DPMs and policy gradient methods.

Following Vignac et al. [61], we consider graphs with categorical node and edge attributes, allowing representation of diverse structured data like molecules. Let $\mathcal{X}$ and $\mathcal{E}$ be the space of categories for nodes and edges, respectively, with cardinalities $a = |\mathcal{X}|$ and $b = |\mathcal{E}|$. For a graph with $n$ nodes, we denote the attribute of node $i$ by a one-hot encoding vector $\boldsymbol{x}^{(i)} \in \mathbb{R}^a$. Similarly, the attribute of the edge[1] from node $i$ to node $j$ is represented as $\boldsymbol{e}^{(ij)} \in \mathbb{R}^b$. By grouping these one-hot vectors, the graph can then be represented as a tuple $\boldsymbol{G} \triangleq (\boldsymbol{X}, \boldsymbol{E})$, where $\boldsymbol{X} \in \mathbb{R}^{n \times a}$ and $\boldsymbol{E} \in \mathbb{R}^{n \times n \times b}$.

## 3.1 Graph Diffusion Probabilistic Models

Graph diffusion probabilistic models (DPMs) [61] involve a forward diffusion process $q(\boldsymbol{G}_{1:T}|\boldsymbol{G}_0) = \prod_{t=1}^T q(\boldsymbol{G}_t|\boldsymbol{G}_{t-1})$, which gradually corrupts a data distribution $q(\boldsymbol{G}_0)$ into a simple noise distribution $q(\boldsymbol{G}_T)$ over a specified number of diffusion steps, denoted as $T$. The transition distribution $q(\boldsymbol{G}_t|\boldsymbol{G}_{t-1})$ can be factorized into a product of categorical distributions for individual nodes and edges, i.e., $q(\boldsymbol{x}_t^{(i)}|\boldsymbol{x}_{t-1}^{(i)})$ and $q(\boldsymbol{e}_t^{(ij)}|\boldsymbol{e}_{t-1}^{(ij)})$. For simplicity, superscripts are omitted when no ambiguity is caused in the following. The transition distribution for each node is defined as $q(\boldsymbol{x}_t|\boldsymbol{x}_{t-1}) = \mathrm{Cat}(\boldsymbol{x}_t; \boldsymbol{x}_{t-1}\boldsymbol{Q}_t)$, where the transition matrix is chosen as $\boldsymbol{Q}_t \triangleq \alpha_t \boldsymbol{I} + (1-\alpha_t)(\mathbf{1}_a \mathbf{1}_a^\top)/a$, with $\alpha_t$ transitioning from 1 to 0 as $t$ increases [2]. It then follows that $q(\boldsymbol{x}_t|\boldsymbol{x}_0) = \mathrm{Cat}(\boldsymbol{x}_t; \boldsymbol{x}_0 \bar{\boldsymbol{Q}}_t)$ and $q(\boldsymbol{x}_{t-1}|\boldsymbol{x}_t, \boldsymbol{x}_0) = \mathrm{Cat}(\boldsymbol{x}_{t-1}; \frac{\boldsymbol{x}_t \boldsymbol{Q}_t^\top \odot \boldsymbol{x}_0 \bar{\boldsymbol{Q}}_{t-1}}{\boldsymbol{x}_0 \bar{\boldsymbol{Q}}_t \boldsymbol{x}_t^\top})$, where $\bar{\boldsymbol{Q}}_t \triangleq \boldsymbol{Q}_1 \boldsymbol{Q}_2 \cdots \boldsymbol{Q}_t$ and $\odot$ denotes element-wise product. The design choice of $\boldsymbol{Q}_t$ ensures that $q(\boldsymbol{x}_T|\boldsymbol{x}_0) \approx \mathrm{Cat}(\boldsymbol{x}_T; \mathbf{1}_a/a)$, i.e., a uniform distribution over $\mathcal{X}$. The transition distribution for edges is defined similarly, and we omit it for brevity.

Given the forward diffusion process, a parametric reverse denoising process $p_\theta(\boldsymbol{G}_{0:T}) = p(\boldsymbol{G}_T) \prod_{t=1}^T p_\theta(\boldsymbol{G}_{t-1}|\boldsymbol{G}_t)$ is then learned to recover the data distribution from $p(\boldsymbol{G}_T) \approx q(\boldsymbol{G}_T)$ (an approximate uniform distribution). The reverse transition $p_\theta(\boldsymbol{G}_{t-1}|\boldsymbol{G}_t)$ is a product of categorical distributions over nodes and edges, denoted as $p_\theta(\boldsymbol{x}_{t-1}|\boldsymbol{G}_t)$ and $p_\theta(\boldsymbol{e}_{t-1}|\boldsymbol{G}_t)$. Notably, in line with the $\boldsymbol{x}_0$-parameterization used in continuous DPMs [25, 32], $p_\theta(\boldsymbol{x}_{t-1}|\boldsymbol{G}_t)$ is modeled as:

$$p_\theta(\boldsymbol{x}_{t-1}|\boldsymbol{G}_t) \triangleq \sum_{\widetilde{\boldsymbol{x}}_0 \in \mathcal{X}} q(\boldsymbol{x}_{t-1}|\boldsymbol{x}_t, \widetilde{\boldsymbol{x}}_0) p_\theta(\widetilde{\boldsymbol{x}}_0|\boldsymbol{G}_t), \tag{1}$$

where $p_\theta(\widetilde{\boldsymbol{x}}_0|\boldsymbol{G}_t)$ is a neural network predicting the posterior probability of $\boldsymbol{x}_0$ given a noisy graph $\boldsymbol{G}_t$. For edges, each definition is analogous and thus omitted.

The model is learned with a graph dataset $\mathcal{D}$ by maximizing the following objective [61]:

$$\mathcal{J}_{\mathrm{GDPM}}(\theta) = \mathbb{E}_{\boldsymbol{G}_0, t} \mathbb{E}_{q(\boldsymbol{G}_t|\boldsymbol{G}_0)} \left[\log p_\theta(\boldsymbol{G}_0|\boldsymbol{G}_t)\right], \tag{2}$$

where $\boldsymbol{G}_0$ and $t$ follow uniform distributions over $\mathcal{D}$ and $[\![1, T]\!]$, respectively. After learning, graph samples can then be generated by first sampling $\boldsymbol{G}_T$ from $p(\boldsymbol{G}_T)$ and subsequently sampling $\boldsymbol{G}_t$ from $p_\theta(\boldsymbol{G}_{t-1}|\boldsymbol{G}_t)$, resulting in a generation trajectory $(\boldsymbol{G}_T, \boldsymbol{G}_{T-1}, \ldots, \boldsymbol{G}_0)$.

## 3.2 Markov Decision Process and Policy Gradient

Markov decision processes (MDPs) are commonly used to model sequential decision-making problems [17]. An MDP is formally defined by a quintuple $(\mathcal{S}, \mathcal{A}, P, r, \rho_0)$, where $\mathcal{S}$ is the state space containing all possible environment states, $\mathcal{A}$ is the action space comprising all available potential actions, $P$ is the transition function determining the probabilities of state transitions, $r$ is the reward signal, and $\rho_0$ gives the distribution of the initial state.

In the context of an MDP, an agent engages with the environment across multiple steps. At each step $t$, the agent observes a state $\boldsymbol{s}_t \in \mathcal{S}$ and selects an action $\boldsymbol{a}_t \in \mathcal{A}$ based on its policy distribution

---

[1]For convenience, "no edge" is treated as a special type of edge.

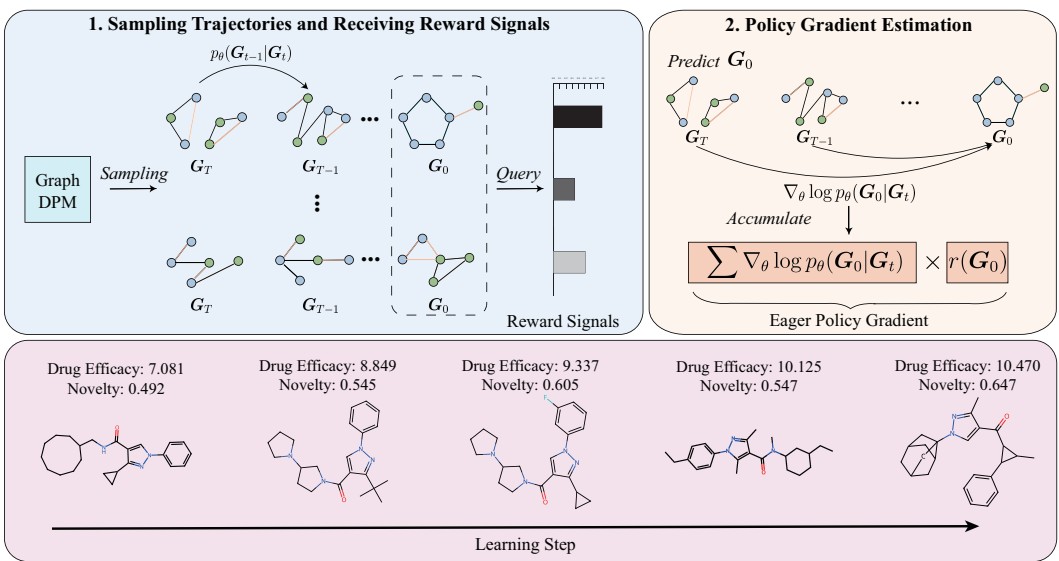

Figure 1: Overview of GDPO. (1) In each optimization step, GDPO samples multiple generation trajectories from the current Graph DPM and queries the reward function with different $G_0$. (2) For each trajectory, GDPO accumulates the gradient $\nabla_\theta \log p_\theta(G_0|G_t)$ of each $(G_0, G_t)$ pair and assigns a weight to the aggregated gradient based on the corresponding reward signal. Finally, GDPO estimates the *eager policy gradient* by averaging the aggregated gradient from all trajectories.

$\pi_\theta(\boldsymbol{a}_t|\boldsymbol{s}_t)$. Subsequently, the agent receives a reward $r(\boldsymbol{s}_t, \boldsymbol{a}_t)$ and transitions to a new state $\boldsymbol{s}_{t+1}$ following the transition function $P(\boldsymbol{s}_{t+1}|\boldsymbol{s}_t, \boldsymbol{a}_t)$. As the agent interacts in the MDP (starting from an initial state $\boldsymbol{s}_0 \sim \rho_0$), it generates a trajectory (i.e., a sequence of states and actions) denoted as $\boldsymbol{\tau} = (\boldsymbol{s}_0, \boldsymbol{a}_0, \boldsymbol{s}_1, \boldsymbol{a}_1, \ldots, \boldsymbol{s}_T, \boldsymbol{a}_T)$. The cumulative reward over a trajectory $\boldsymbol{\tau}$ is given by $R(\boldsymbol{\tau}) = \sum_{t=0}^{T} r(\boldsymbol{s}_t, \boldsymbol{a}_t)$. In most scenarios, the objective is to maximize the following expectation:

$$\mathcal{J}_{\text{RL}}(\theta) = \mathbb{E}_{\boldsymbol{\tau} \sim p(\boldsymbol{\tau}|\pi_\theta)} \left[ R(\boldsymbol{\tau}) \right]. \tag{3}$$

Policy gradient methods aim to estimate $\nabla_\theta \mathcal{J}_{\text{RL}}(\theta)$ and thus solve the problem by gradient descent. An important result is the policy gradient theorem [19], which estimates $\nabla_\theta \mathcal{J}_{\text{RL}}(\theta)$ as follows:

$$\nabla_\theta \mathcal{J}_{\text{RL}}(\theta) = \mathbb{E}_{\boldsymbol{\tau} \sim p(\boldsymbol{\tau}|\pi_\theta)} \left[ \sum_{t=0}^{T} \nabla_\theta \log \pi_\theta(\boldsymbol{a}_t|\boldsymbol{s}_t) R(\boldsymbol{\tau}) \right]. \tag{4}$$

The REINFORCE algorithm [58] provides a simple method for estimating the above policy gradient using Monte-Carlo simulation, which will be adopted and discussed in the following section.

## 4 Method

In this section, we study the problem of learning graph DPMs from arbitrary reward signals. We first present an MDP formulation of the problem and conduct an analysis on the failure of a direct application of REINFORCE. Based on the analysis, we introduce a substitute termed *eager policy gradient*, which forms the core of our method *Graph Diffusion Policy Optimization* (GDPO).

### 4.1 A Markov Decision Process Formulation

A graph DPM defines a sample distribution $p_\theta(G_0)$ through its reverse denoising process $p_\theta(G_{0:T})$. Given a reward signal $r(\cdot)$ for $G_0$, we aim to maximize the expected reward (ER) over $p_\theta(G_0)$:

$$\mathcal{J}_{\text{ER}}(\theta) = \mathbb{E}_{G_0 \sim p_\theta(G_0)} \left[ r(G_0) \right]. \tag{5}$$

However, directly optimizing $\mathcal{J}_{\text{ER}}(\theta)$ is challenging since the likelihood $p_\theta(G_0)$ is unavailable [25] and $r(\cdot)$ is black-box, hindering the use of typical RL algorithms [6]. Following Fan et al. [16], we

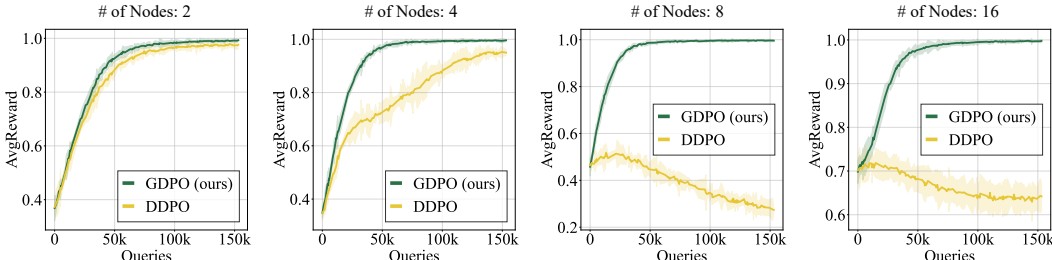

Figure 2: Toy experiment comparing DDPO and GDPO. We generate connected graphs with increasing number of nodes. Node categories are disregarded, and the edge categories are binary, indicating whether two nodes are linked. The graph DPM is initialized randomly as a one-layer graph transformer from DiGress [61]. The diffusion step $T$ is set to 50, and the reward signal $r(\boldsymbol{G}_0)$ is defined as 1 if $\boldsymbol{G}_0$ is connected and 0 otherwise. We use 256 trajectories for gradient estimation in each update. The learning curve illustrates the diminishing performance of DDPO as the number of nodes increases, while GDPO consistently performs well.

formulate the denoising process as a $T$-step MDP and obtain an equivalent objective. Using notations in Sec. 3, we define the MDP of graph DPMs as follows:

$$\boldsymbol{s}_t \triangleq (\boldsymbol{G}_{T-t}, T-t), \quad \boldsymbol{a}_t \triangleq \boldsymbol{G}_{T-t-1}, \quad \pi_\theta(\boldsymbol{a}_t|\boldsymbol{s}_t) \triangleq p_\theta(\boldsymbol{G}_{T-t-1}|\boldsymbol{G}_{T-t}),$$

$$P(\boldsymbol{s}_{t+1}|\boldsymbol{s}_t,\boldsymbol{a}_t) \triangleq (\delta_{\boldsymbol{G}_{T-t-1}}, \delta_{T-t-1}), \quad r(\boldsymbol{s}_t,\boldsymbol{a}_t) \triangleq r(\boldsymbol{G}_0) \text{ if } t=T, \; r(\boldsymbol{s}_t,\boldsymbol{a}_t) \triangleq 0 \text{ if } t < T,$$

(6)

where the initial state $\boldsymbol{s}_0$ corresponds to the initial noisy graph $\boldsymbol{G}_T$ and the policy corresponds to the reverse transition distribution. As a result, the graph generation trajectory $(\boldsymbol{G}_T, \boldsymbol{G}_{T-1}, \ldots, \boldsymbol{G}_0)$ can be considered as a state-action trajectory $\boldsymbol{\tau}$ produced by an agent acting in the MDP. It then follows that $p(\boldsymbol{\tau}|\pi_\theta) = p_\theta(\boldsymbol{G}_{0:T})$.[2] Moreover, we have $R(\boldsymbol{\tau}) = \sum_{t=0}^{T} r(\boldsymbol{s}_t, \boldsymbol{a}_t) = r(\boldsymbol{G}_0)$. Therefore, the expected cumulative reward of the agent $\mathcal{J}_{\mathrm{RL}}(\theta) = \mathbb{E}_{p(\boldsymbol{\tau}|\pi_\theta)}[R(\boldsymbol{\tau})] = \mathbb{E}_{p_\theta(\boldsymbol{G}_{0:T})}[r(\boldsymbol{G}_0)]$ is equivalent to $\mathcal{J}_{\mathrm{ER}}(\theta)$, and thus $\mathcal{J}_{\mathrm{ER}}(\theta)$ can also be optimized with the policy gradient $\nabla_\theta \mathcal{J}_{\mathrm{RL}}(\theta)$:

$$\nabla_\theta \mathcal{J}_{\mathrm{RL}}(\theta) = \mathbb{E}_{\boldsymbol{\tau}} \left[ r(\boldsymbol{G}_0) \sum_{t=1}^{T} \nabla_\theta \log p_\theta(\boldsymbol{G}_{t-1}|\boldsymbol{G}_t) \right],$$

(7)

where the generation trajectory $\boldsymbol{\tau}$ follows the parametric reverse process $p_\theta(\boldsymbol{G}_{0:T})$.

## 4.2 Learning Graph DPMs with Policy Gradient

The policy gradient $\nabla_\theta \mathcal{J}_{\mathrm{RL}}(\theta)$ in Eq. (7) is generally intractable and an efficient estimation is necessary. In a related setting centered on continuous-variable DPMs for image generation, DDPO [6] estimates the policy gradient $\nabla_\theta \mathcal{J}_{\mathrm{RL}}(\theta)$ with REINFORCE and achieves great results. This motivates us to also try REINFORCE on graph DPMs, i.e., to approximate Eq. (7) with a Monte Carlo estimation:

$$\nabla_\theta \mathcal{J}_{\mathrm{RL}} \approx \frac{1}{K} \sum_{k=1}^{K} \frac{T}{|\mathcal{T}_k|} \sum_{t \in \mathcal{T}_k} r(\boldsymbol{G}_0^{(k)}) \nabla_\theta \log p_\theta(\boldsymbol{G}_{t-1}^{(k)}|\boldsymbol{G}_t^{(k)}),$$

(8)

where $\{\boldsymbol{G}_{0:T}^{(k)}\}_{k=1}^{K}$ are $K$ trajectories sampled from $p_\theta(\boldsymbol{G}_{0:T})$ and $\{\mathcal{T}_k \subset [\![1, T]\!]\}_{k=1}^{K}$ are uniformly random subsets of timesteps (which avoid summing over all timesteps and accelerate the estimation).

However, we empirically observe that it rarely converges on graph DPMs. To investigate this, we design a toy experiment, where the reward signal is whether $\boldsymbol{G}_0$ is connected. The graph DPMs are randomly initialized and optimized using Eq. (8). We refer to this setting as DDPO. Fig. 2 depicts the learning curves, where the horizontal axis represents the number of queries to the reward signal and the vertical axis represents the average reward. The results demonstrate that DDPO fails to converge to a high reward signal area when generating graphs with more than 4 nodes. Furthermore, as the

---

[2]With a slight abuse of notation we will use $\boldsymbol{\tau} = \boldsymbol{G}_{0:T}$ and $\boldsymbol{\tau} = (\boldsymbol{s}_0, \boldsymbol{a}_0, \boldsymbol{s}_1, \boldsymbol{a}_1, \ldots, \boldsymbol{s}_T, \boldsymbol{a}_T)$ interchangeably, which should not confuse as the MDP relates them with a bijection.

number of nodes increases, the fluctuation of the learning curves grows significantly. This implies that DDPO is essentially unable to optimize properly on randomly initialized models. We conjecture that the failure is due to the vast space constituted by discrete graph trajectories and the well-known high variance issue of REINFORCE [58]. A straightforward method to reduce variance is to sample more trajectories. However, this is typically expensive in DPMs, as each trajectory requires multiple rounds of model inference. Moreover, evaluating the reward signals of additional trajectories also incurs high computational costs, such as drug simulation [48].

This prompts us to delve deeper at a micro level. Since the policy gradient estimation in Eq. (8) is a weighted summation of gradients, we first inspect each summand term $\nabla_\theta \log p_\theta(\boldsymbol{G}_{t-1}|\boldsymbol{G}_t)$. With the parameterization Eq. (1) described in Sec. 3.1, it has the following form:

$$\nabla_\theta \log p_\theta(\boldsymbol{G}_{t-1}|\boldsymbol{G}_t) = \frac{1}{p_\theta(\boldsymbol{G}_{t-1}|\boldsymbol{G}_t)} \sum_{\widetilde{\boldsymbol{G}}_0} \underbrace{q(\boldsymbol{G}_{t-1}|\boldsymbol{G}_t, \widetilde{\boldsymbol{G}}_0)}_{\text{weight}} \underbrace{\nabla_\theta p_\theta(\widetilde{\boldsymbol{G}}_0|\boldsymbol{G}_t)}_{\text{gradient}}, \tag{9}$$

where we can view the "weight" term as a weight assigned to the gradient $\nabla_\theta p_\theta(\widetilde{\boldsymbol{G}}_0|\boldsymbol{G}_t)$, and thus $\nabla_\theta \log p_\theta(\boldsymbol{G}_{t-1}|\boldsymbol{G}_t)$ as a weighted sum of such gradients, with $\widetilde{\boldsymbol{G}}_0$ taken over all possible graphs. Intuitively, the gradient $\nabla_\theta p_\theta(\widetilde{\boldsymbol{G}}_0|\boldsymbol{G}_t)$ promotes the probability of predicting $\widetilde{\boldsymbol{G}}_0$ from $\boldsymbol{G}_t$. Note, however, that the weight $q(\boldsymbol{G}_{t-1}|\boldsymbol{G}_t, \widetilde{\boldsymbol{G}}_0)$ is completely independent of $r(\widetilde{\boldsymbol{G}}_0)$ and could assign large weight for $\widetilde{\boldsymbol{G}}_0$ that has low reward. Since the weighted sum in Eq. (9) can be dominated by gradient terms with large $q(\boldsymbol{G}_{t-1}|\boldsymbol{G}_t, \widetilde{\boldsymbol{G}}_0)$, given a particular sampled trajectory, it is fairly possible that $\nabla_\theta \log p_\theta(\boldsymbol{G}_{t-1}|\boldsymbol{G}_t)$ increases the probabilities of predicting undesired $\widetilde{\boldsymbol{G}}_0$ with low rewards from $\boldsymbol{G}_t$. This explains why Eq. (8) tends to produce fluctuating and unreliable policy gradient estimates when the number of Monte Carlo samples (i.e., $K$ and $|\mathcal{T}_k|$) is limited. To further analyze why DDPO does not yield satisfactory results, we present additional findings in Appendix A.5. Besides, we discuss the impact of importance sampling techniques in the same section.

### 4.3 Graph Diffusion Policy Optimization

To address the above issues, we suggest a slight modification to Eq. (8) and obtain a new policy gradient denoted as $\boldsymbol{g}(\theta)$:

$$\boldsymbol{g}(\theta) \triangleq \frac{1}{K} \sum_{k=1}^{K} \frac{T}{|\mathcal{T}_k|} \sum_{t \in \mathcal{T}_k} r(\boldsymbol{G}_0^{(k)}) \nabla_\theta \log p_\theta(\boldsymbol{G}_0^{(k)}|\boldsymbol{G}_t^{(k)}), \tag{10}$$

which we refer to as the *eager policy gradient*. Intuitively, although the number of possible graph trajectories is tremendous, if we partition them into different equivalence classes according to $\boldsymbol{G}_0$, where trajectories with the same $\boldsymbol{G}_0$ are considered equivalent, then the number of these equivalence classes will be much smaller than the number of graph trajectories. The optimization over these equivalence classes will be much easier than optimizing in the entire trajectory space.

Technically, by replacing the summand gradient term $\nabla_\theta \log p_\theta(\boldsymbol{G}_{t-1}|\boldsymbol{G}_t)$ with $\nabla_\theta \log p_\theta(\boldsymbol{G}_0|\boldsymbol{G}_t)$ in Eq. (8), we skip the weighted sum in Eq. (9) and directly promotes the probability of predicting $\boldsymbol{G}_0$ which has higher reward from $\boldsymbol{G}_t$ at all timestep $t$. As a result, our estimation does not focus on how $\boldsymbol{G}_t$ changes to $\boldsymbol{G}_{t-1}$ within the trajectory; instead, it aims to force the model's generated results to be close to the desired $\boldsymbol{G}_0$, which can be seen as optimizing in equivalence classes. While being a biased estimator of the policy gradient $\nabla_\theta \mathcal{J}_{\text{RL}}(\theta)$, the eager policy gradient consistently leads to more stable learning and better performance than DDPO, as demonstrated in Fig. 2. We present the resulting method in Fig. 1 and Algorithm 1, naming it *Graph Diffusion Policy Optimization* (GDPO).

## 5 Reward Functions for Graph Generation

In this work, we study both general graph and molecule reward signals that are crucial in real-world tasks. Below, we elaborate on how we formulate diverse reward signals as numerical functions.

### 5.1 Reward Functions for General Graph Generation

**Validity.** For graph generation, a common objective is to generate a specific type of graph. For instance, one might be interested in graphs that can be drawn without edges crossing each other [43].

Table 1: General graph generation on SBM and Planar datasets.

| Method | Planar Graphs | | | | SBM Graphs | | | |
|---|---|---|---|---|---|---|---|---|
| | $Deg \downarrow$ | $Clus \downarrow$ | $Orb \downarrow$ | $V.U.N\ (\%) \uparrow$ | $Deg \downarrow$ | $Clus \downarrow$ | $Orb \downarrow$ | $V.U.N\ (\%) \uparrow$ |
| GraphRNN | $24.51 \pm 3.22$ | $9.03 \pm 0.78$ | $2508.30 \pm 30.81$ | $0$ | $6.92 \pm 1.13$ | $1.72 \pm 0.05$ | $3.15 \pm 0.23$ | $4.92 \pm 0.35$ |
| SPECTRE | $2.55 \pm 0.34$ | $2.52 \pm 0.26$ | $2.42 \pm 0.37$ | $25.46 \pm 1.33$ | $1.92 \pm 1.21$ | $1.64 \pm 0.06$ | $1.67 \pm 0.14$ | $53.76 \pm 3.62$ |
| GDSS | $10.81 \pm 0.86$ | $12.99 \pm 0.22$ | $38.71 \pm 0.83$ | $0.78 \pm 0.72$ | $15.53 \pm 1.30$ | $3.50 \pm 0.81$ | $15.98 \pm 2.30$ | $0$ |
| MOOD | $5.73 \pm 0.82$ | $11.87 \pm 0.34$ | $30.62 \pm 0.67$ | $1.21 \pm 0.83$ | $12.87 \pm 1.20$ | $3.06 \pm 0.37$ | $2.81 \pm 0.35$ | $0$ |
| DiGress | $1.43 \pm 0.90$ | $1.22 \pm 0.32$ | $1.72 \pm 0.44$ | $70.02 \pm 2.17$ | $1.63 \pm 1.51$ | $1.50 \pm 0.04$ | $1.70 \pm 0.16$ | $60.94 \pm 4.98$ |
| DDPO | $109.59 \pm 36.69$ | $31.47 \pm 4.96$ | $504.19 \pm 17.61$ | $2.34 \pm 1.10$ | $250.06 \pm 7.44$ | $2.93 \pm 0.32$ | $6.65 \pm 0.45$ | $31.25 \pm 5.22$ |
| GDPO (ours) | $\mathbf{0.03} \pm 0.04$ | $\mathbf{0.62} \pm 0.11$ | $\mathbf{0.02} \pm 0.01$ | $\mathbf{73.83} \pm 2.49$ | $\mathbf{0.15} \pm 0.13$ | $\mathbf{1.50} \pm 0.01$ | $\mathbf{1.12} \pm 0.14$ | $\mathbf{80.08} \pm 2.07$ |

For such objectives, the reward function $r_{\mathrm{val}}(\cdot)$ is then formulated as binary, with $r_{\mathrm{val}}(\boldsymbol{G}_0) \triangleq 1$ indicating that the generated graph $\boldsymbol{G}_0$ conforms to the specified type; otherwise, $r_{\mathrm{val}}(\boldsymbol{G}_0) \triangleq 0$.

**Similarity.** In certain scenarios, the objective is to generate graphs that resemble a known set of graphs $\mathcal{D}$ at the distribution level, based on a pre-defined distance metric $d(\cdot, \cdot)$ between sets of graphs. As an example, the $Deg(\mathcal{G}, \mathcal{D})$ [38] measures the maximum mean discrepancy (MMD) [18] between the degree distributions of a set $\mathcal{G}$ of generated graphs and the given graphs $\mathcal{D}$. Since our method requires a reward for each single generated graph $\boldsymbol{G}_0$, we simply adopt $Deg(\{\boldsymbol{G}_0\}, \mathcal{D})$ as the signal. As the magnitude of reward is critical for policy gradients [58], we define $r_{\mathrm{deg}}(\boldsymbol{G}_0) \triangleq \exp\left(-Deg(\{\boldsymbol{G}_0\}, \mathcal{D})^2/\sigma^2\right)$, where the $\sigma$ controls the reward distribution, ensuring that the reward lies within the range of 0 to 1. The other two similar distance metrics are $Clus(\mathcal{G}, \mathcal{D})$ and $Orb(\mathcal{G}, \mathcal{D})$, which respectively measure the distances between two sets of graphs in terms of the distribution of clustering coefficients [55] and the distribution of substructures [1]. Based on the two metrics, we define two reward signals analogous to $r_{\mathrm{deg}}$, namely, $r_{\mathrm{clus}}$ and $r_{\mathrm{orb}}$.

## 5.2 Reward Functions for Molecular Graph Generation

**Novelty.** A primary objective of molecular graph generation is to discover novel drugs with desired therapeutic potentials. Due to drug patent restrictions, the novelty of generated molecules has paramount importance. The Tanimoto similarity [4], denoted as $J(\cdot, \cdot)$, measures the chemical similarity between two molecules, defined by the Jaccard index of molecule fingerprint bits. Specifically, $J \in [0, 1]$, and $J(\boldsymbol{G}_0, \boldsymbol{G}_0') = 1$ indicates that two molecules $\boldsymbol{G}_0$ and $\boldsymbol{G}_0'$ have identical fingerprints. Following Xie et al. [65], we define the novelty of a generated graph $\boldsymbol{G}_0$ as $NOV(\boldsymbol{G}_0) \triangleq 1 - \max_{\boldsymbol{G}_0' \in \mathcal{D}} J(\boldsymbol{G}_0, \boldsymbol{G}_0')$, i.e., the similarity gap between $\boldsymbol{G}_0$ and its nearest neighbor in the training dataset $\mathcal{D}$, and further define $r_{\mathrm{NOV}}(\boldsymbol{G}_0) \triangleq NOV(\boldsymbol{G}_0)$.

**Drug-Likeness**. Regarding the efficacy of molecular graph generation in drug design, a critical indicator is the binding affinity between the generated drug candidate and a target protein. The docking score [10], denoted as $DS(\cdot)$, estimates the binding energy (in kcal/mol) between the ligand and the target protein through physical simulations in 3D space. Following Lee et al. [37], we clip the docking score in the range $[-20, 0]$ and define the reward function as $r_{\mathrm{DS}}(\boldsymbol{G}_0) \triangleq -DS(\boldsymbol{G}_0)/20$.

Another metric is the quantitative estimate of drug-likeness $QED(\cdot)$, which measures the chemical properties to gauge the likelihood of a molecule being a successful drug [5]. As it takes values in the range $[0, 1]$, we adopt $r_{\mathrm{QED}}(\boldsymbol{G}_0) \triangleq \mathbb{I}[QED(\boldsymbol{G}_0) > 0.5]$.

**Synthetic Accessibility**. The synthetic accessibility [7] $SA(\cdot)$ evaluates the inherent difficulty in synthesizing a chemical compound, with values in the range from 1 to 10. We follow Lee et al. [37] and use a normalized version as the reward function: $r_{\mathrm{SA}}(\boldsymbol{G}_0) \triangleq (10 - SA(\boldsymbol{G}_0))/9$.

## 6 Experiments

In this section, we first examine the performance of GDPO on both general graph generation tasks and molecular graph generation tasks. Then, we conduct several ablation studies to investigate the effectiveness of GDPO's design. Our code can be found in the supplementary material.

Table 2: Molecule property optimization results on ZINC250k.

| Method | Metric | Target Protein | | | | |
|---|---|---|---|---|---|---|
| | | *parp1* | *fa7* | *5ht1b* | *braf* | *jak2* |
| GCPN | *Hit Ratio* | 0 | 0 | 1.455 ± 1.173 | 0 | 0 |
| | *DS (top 5%)* | -8.102± 0.105 | -6.688±0.186 | -8.544± 0.505 | -8.713± 0.155 | -8.073±0.093 |
| REINVENT | *Hit Ratio* | 0.480 ± 0.344 | 0.213 ± 0.081 | 2.453 ± 0.561 | 0.127 ± 0.088 | 0.613 ± 0.167 |
| | *DS (top 5%)* | -8.702 ± 0.523 | -7.205 ± 0.264 | -8.770 ± 0.316 | -8.392 ± 0.400 | -8.165 ± 0.277 |
| FREED | *Hit Ratio* | 4.627 ± 0.727 | 1.332 ± 0.113 | 16.767 ± 0.897 | 2.940 ± 0.359 | 5.800 ± 0.295 |
| | *DS (top 5%)* | -10.579 ± 0.104 | -8.378 ± 0.044 | -10.714 ± 0.183 | -10.561 ± 0.080 | -9.735 ± 0.022 |
| MOOD | *Hit Ratio* | 7.017 ± 0.428 | 0.733 ± 0.141 | 18.673 ± 0.423 | 5.240 ± 0.285 | 9.200 ± 0.524 |
| | *DS (top 5%)* | -10.865 ± 0.113 | -8.160 ± 0.071 | -11.145 ± 0.042 | -11.063 ± 0.034 | -10.147 ± 0.060 |
| DiGress | *Hit Ratio* | 0.366 ± 0.146 | 0.182 ± 0.232 | 4.236 ± 0.887 | 0.122 ± 0.141 | 0.861 ± 0.332 |
| | *DS (top 5%)* | -9.219 ± 0.078 | -7.736 ± 0.156 | -9.280 ± 0.198 | -9.052 ± 0.044 | -8.706 ± 0.222 |
| DiGress-guidance | *Hit Ratio* | 1.172±0.672 | 0.321±0.370 | 2.821± 1.140 | 0.152±0.303 | 0.311±0.621 |
| | *DS (top 5%)* | -9.463± 0.524 | -7.318±0.213 | -8.971± 0.395 | -8.825± 0.459 | -8.360±0.217 |
| DDPO | *Hit Ratio* | 0.419 ± 0.280 | 0.342 ± 0.685 | 5.488 ± 1.989 | 0.445 ± 0.297 | 1.717 ± 0.684 |
| | *DS (top 5%)* | -9.247 ± 0.242 | -7.739 ± 0.244 | -9.488 ± 0.287 | -9.470 ± 0.373 | -8.990 ± 0.221 |
| GDPO (ours) | *Hit Ratio* | **9.814** ± 1.352 | **3.449** ± 0.188 | **34.359** ± 2.734 | **9.039** ± 1.473 | **13.405** ± 1.515 |
| | *DS (top 5%)* | **-10.938** ± 0.042 | **-8.691** ± 0.074 | **-11.304** ± 0.093 | **-11.197** ± 0.132 | **-10.183** ± 0.124 |

### 6.1 General Graph Generation

**Datasets and Baselines.** Following DiGress [61], we evaluate GDPO on two benchmark datasets: SBM (200 nodes) and Planar (64 nodes), each consisting of 200 graphs. We compare GDPO with GraphRNN [69], SPECTRE [43], GDSS [31], MOOD [37] and DiGress. The first two models are based on RNN and GAN, respectively. The remaining methods are graph DPMs, and MOOD employs an additional property predictor. We also test DDPO [6], i.e., graph DPMs optimized with Eq. (8).

**Implementation.** We set $T = 1000$, $|\mathcal{T}| = 200$, and $N = 100$. The number of trajectory samples $K$ is 64 for SBM and 256 for Planar. We use a DiGress model with 10 layers. More implementation details can be found in Appendix A.1.

**Metrics and Reward Functions.** We consider four metrics: $Deg(\mathcal{G}, \mathcal{D}_{test})$, $Clus(\mathcal{G}, \mathcal{D}_{test})$, $Orb(\mathcal{G}, \mathcal{D}_{test})$, and the *V.U.N* metrics. *V.U.N* measures the proportion of generated graphs that are valid, unique, and novel. The reward function is defined as follows:

$$r_{\text{general}} = 0.1 \times (r_{\text{deg}} + r_{\text{clus}} + r_{\text{orb}}) + 0.7 \times r_{\text{val}}, \tag{11}$$

where we do not explicitly incorporate uniqueness and novelty. All rewards are calculated on the training dataset if a reference graph set is required. All evaluation metrics are calculated on the test dataset. More details about baselines, reward signals, and metrics are in Appendix A.3.

**Results.** Table 1 summarizes GDPO's superior performance in general graph generation, showing notable improvements in *Deg* and *V.U.N* across both SBM and Planar datasets. On the Planar dataset, GDPO significantly reduces distribution distance, achieving an $81.97\%$ average decrease in metrics of *Deg*, *Clus*, and *Orb* compared to DiGress (the best baseline method). For the SBM dataset, GDPO has a $41.64\%$ average improvement. The low distributional distances to the test dataset suggests that GDPO accurately captures the data distribution with well-designed rewards. Moreover, we observe that our method outperforms DDPO by a large margin, primarily because the graphs in Planar and SBM contain too many nodes, which aligns with the observation in Fig. 2.

### 6.2 Molecule Property Optimization

**Datasets and Baselines.** Molecule property optimization aims to generate molecules with desired properties. We evaluate our method on two large molecule datasets: ZINC250k [27] and MOSES [49]. The ZINC250k dataset comprises 249,456 molecules, each containing 9 types of atoms, with a maximum node count of 38; the MOSES dataset consists of 1,584,663 molecules, with 8 types of atoms and a maximum node count of 30. We compare GDPO with several leading methods:

Table 3: Molecule property optimization results on MOSES.

| Method | Metric | Target Protein | | | | |
|--------|--------|-------|------|-------|------|------|
| | | *parp1* | *fa7* | *5ht1b* | *braf* | *jak2* |
| FREED | *Hit Ratio* | $0.532 \pm 0.614$ | $0$ | $4.255 \pm 0.869$ | $0.263 \pm 0.532$ | $0.798 \pm 0.532$ |
| | *DS (top 5%)* | $-9.313 \pm 0.357$ | $-7.825 \pm 0.167$ | $-9.506 \pm 0.236$ | $-9.306 \pm 0.327$ | $-8.594 \pm 0.240$ |
| MOOD | *Hit Ratio* | $5.402 \pm 0.042$ | $0.365 \pm 0.200$ | $26.143 \pm 1.647$ | $3.932 \pm 1.290$ | $11.301 \pm 1.154$ |
| | *DS (top 5%)* | $-9.814 \pm 1.352$ | $-7.974 \pm 0.029$ | $10.734 \pm 0.049$ | $-10.722 \pm 0.135$ | $-10.158 \pm 0.185$ |
| DiGress | *Hit Ratio* | $0.231 \pm 0.463$ | $0.113 \pm 0.131$ | $3.852 \pm 5.013$ | $0$ | $0.228 \pm 0.457$ |
| | *DS (top 5%)* | $-9.223 \pm 0.083$ | $-6.644 \pm 0.533$ | $-8.640 \pm 0.907$ | $8.522 \pm 1.017$ | $-7.424 \pm 0.994$ |
| DDPO | *Hit Ratio* | $3.037 \pm 2.107$ | $0.504 \pm 0.667$ | $7.855 \pm 1.745$ | $0$ | $3.943 \pm 2.204$ |
| | *DS (top 5%)* | $-9.727 \pm 0.529$ | $-8.025 \pm 0.253$ | $-9.631 \pm 0.123$ | $-9.407 \pm 0.125$ | $-9.404 \pm 0.319$ |
| GDPO (ours) | *Hit Ratio* | $\mathbf{24.711} \pm 1.775$ | $\mathbf{1.393} \pm 0.982$ | $17.646 \pm 2.484$ | $\mathbf{19.968} \pm 2.309$ | $\mathbf{26.688} \pm 2.401$ |
| | *DS (top 5%)* | $\mathbf{-11.002} \pm 0.056$ | $\mathbf{-8.468} \pm 0.058$ | $\mathbf{-10.990} \pm 0.334$ | $\mathbf{-11.337} \pm 0.137$ | $\mathbf{-10.290} \pm 0.069$ |

GCPN [68], REINVENT [47], FREED [67] and MOOD [37]. GCPN, REINVENT and FREED are RL methods that search in the chemical environment. MOOD, based on graph DPMs, employs a property predictor for guided sampling. Similar to general graph generation, we also compare our method with DiGress and DDPO. Besides, we show the performance of DiGress with property predictors, termed as DiGress-guidance.

**Implementation.** We set $T = 500$, $|\mathcal{T}| = 100$, $N = 100$, and $K = 256$ for both datasets. We use the same model structure with DiGress. See more details in Appendix A.1.

**Metrics and Reward Functions.** Following MOOD, we consider two metrics essential for real-world novel drug discovery: **Novel hit ratio (%)** and **Novel top** $5\%$ **docking score**, denoted as *Hit Ratio* and *DS (top 5%)*, respectively. Using the notations from Sec. 5.2, the *Hit Ratio* is the proportion of unique generated molecules that satisfy: *DS* < median *DS* of the known effective molecules, *NOV* > 0.6, *QED* > 0.5, and *SA* < 5. The *DS (top 5%)* is the average *DS* of the top $5\%$ molecules (ranked by *DS*) that satisfy: *NOV* > 0.6, *QED* > 0.5, and *SA* < 5. Since calculating *DS* requires specifying a target protein, we set five different protein targets to fully test GDPO: *parp1*, *fa7*, *5ht1b*, *braf*, and *jak2*. The reward function for molecule property optimization is defined as follows:

$$r_{\mathrm{molecule}} = 0.1 \times (r_{\mathrm{QED}} + r_{\mathrm{SA}}) + 0.3 \times r_{\mathrm{NOV}} + 0.5 \times r_{\mathrm{DS}}. \tag{12}$$

We do not directly use *Hit Ratio* and *DS (top 5%)* as rewards in consideration of method generality. The reward weights are determined through several rounds of search, and we find that assigning a high weight to $r_{\mathrm{NOV}}$ leads to training instability, which is discussed in Sec. 6.3. More details about the experiment settings are discussed in Appendix A.4.

**Results.** In Table 2, GDPO shows significant improvement on ZINC250k, especially in the *Hit Ratio*. A higher *Hit Ratio* means the model is more likely to generate valuable new drugs, and GDPO averagely improves the *Hit Ratio* by $5.72\%$ in comparison with other SOTA methods. For *DS (top 5%)*, GDPO also has a $1.48\%$ improvement on average. Discovering new drugs on MOSES is much more challenging than on ZINC250k due to its vast training dataset. In Table 3, GDPO also shows promising results on MOSES. Despite a less favorable *Hit Ratio* on *5ht1b*, GDPO achieves an average improvement of $12.94\%$ on the other four target proteins. For *DS (top 5%)*, GDPO records an average improvement of $5.54\%$ compared to MOOD, showing a big improvement in drug efficacy.

### 6.3 Generalizability, Sample Efficiency, and A Failure Case

To validate whether GDPO correctly optimizes the model, we test the performance of GDPO on metrics not used in the reward signal. In Table 4, we evaluate the performance on Spectral MMD [43], where the GDPO is optimized by Eq. (11). The results demonstrate that GDPO does

Table 4: Generalizability of GDPO on Spectral MMD.

| Dataset | Methods | | |
|---------|---------|------|------|
| | *DiGress* | *DDPO* | *GDPO (ours)* |
| PLANAR | $1.0353 \pm 0.4474$ | $20.1431 \pm 3.5810$ | $\mathbf{0.8047} \pm 0.2030$ |
| SBM | $1.2024 \pm 0.2874$ | $13.2773 \pm 1.4233$ | $\mathbf{1.0861} \pm 0.2551$ |

not show overfitting; instead, it finds a more powerful model. The results presented in Appendix A.5 further support that GDPO can attain high sample novelty and diversity.

We then investigate two crucial factors for GDPO: 1) the number of trajectories; 2) the selection of the reward signals. We test our method on ZINC250k and set the target proteins as *5ht1b*. In Fig. 3 (a), the results indicate that GDPO exhibits good sampling efficiency, as it achieves a significant improvement in average reward by querying only 10k molecule reward signals, which is much less than the number of molecules contained in ZINC250k. Moreover, the sample efficiency can be further improved by reducing the number of trajectories, but this may lead to training instability. To achieve consistent results, we use 256 trajectories. In Fig. 3 (b), we illustrate a failure case of GDPO when assigning a high weight to $r_{\text{NOV}}$. Gen-

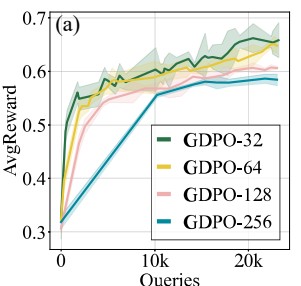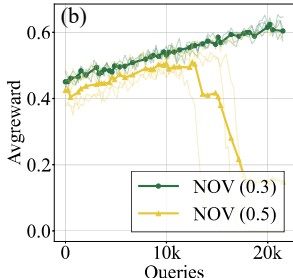

Figure 3: We investigate two key factors of GDPO on ZINC250k, with the target protein being *5ht1b*. Similarly, the vertical axis represents the total queries, while the horizontal axis represents the average reward.(a) We vary the number of trajectories for gradient estimation. (b) We fix the weight of $r_{\text{QED}}$ and $r_{\text{SA}}$, and change the weight of $r_{\text{NOV}}$ while ensuring the total weight is 1.

erating novel samples is challenging. MOOD [37] addresses this challenge by controlling noise in the sampling process, whereas we achieve it by novelty optimization. However, assigning a large weight to $r_{\text{NOV}}$ can lead the model to rapidly degenerate. One potential solution is to gradually increase the weight and conduct multi-stage optimization.

# 7 Conclusion

We introduce GDPO, a novel policy gradient method for learning graph DPMs that effectively addresses the problem of graph generation under given objectives. Evaluation results on both general and molecular graphs indicate that GDPO is compatible with complex multi-objective optimization and achieves state-of-the-art performance on a series of representative graph generation tasks. We discuss some limitations of our work in Appendix A.2. Our future work will investigate the theoretical gap between GDPO and DDPO in order to obtain effective unbiased estimators.

# Acknowledgment

This work is supported by the Zhejiang Provincial Natural Science Foundation of China (LD24F020011) and "Pioneer and Leading Goose" R&D Program of Zhejiang (2024C01167). Chongxuan Li was supported by Beijing Natural Science Foundation (L247030); Beijing Nova Program (No. 20230484416).

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

# A Experimental Details and Additional Results

## A.1 Implementation Details.

For all experiments, we use the graph transformer proposed in DiGress [61] as the graph DPMs, and the models are pre-trained on the training dataset before applying GDPO or DDPO. During fine-tuning, we keep all layers fixed except for attention, set the learning rate to 0.00001, and utilize gradient clipping to limit the gradient norm to be less than or equal to 1. In addition, due to significant numerical fluctuations during reward normalization, we follow DDPO [6] in constraining the normalized reward to the range from $[-5, 5]$. This means that gradients resulting from rewards beyond this range will not contribute to model updates. When there is insufficient memory to generate enough trajectories, we use gradient accumulation to increase the number of trajectories used for gradient estimation. We conducted all experiments on a single A100 GPU with 40GB of VRAM and an AMD EPYC 7352 24-core Processor.

**Training time and efficiency.** Training DiGress on the ZINC250k dataset using a single A100 GPU typically takes 48-72 hours, whereas fine-tuning with GDPO takes only 10 hours (excluding the time for reward function computation). This high efficiency is in line with the findings in the practice of DDPO, which is different from traditional RL methods. Additionally, as in Fig. 3 and Sec 6.3, GDPO effectively improves the average reward of the model using only 10,000 queries. This sample size is notably small compared to the 250,000 samples present in the ZINC250k dataset, showing the impressive sample efficiency of GDPO.

## A.2 Limitations and Broader Impact.

Below we list some limitations of the current work:

- Potential for overoptimization: As an RL-based approach, a recognized limitation is the risk of overoptimization, where the DPM distribution may collapse or diverge excessively from the original distribution. In Section 6.3, we demonstrated a failure case where, with a high weight on novelty in the reward function, GDPO encounters a sudden drop in reward after a period of optimization. Future research could explore the application of regularization techniques, similar to those utilized in recent works such as DPO [51], to mitigate this risk.

- Inherited limitations of DPMs: Our method inherits certain limitations inherent to diffusion models, particularly concerning their training and inference costs. As we do not modify the underlying model architecture, these constraints persist.

- Scalability to large graphs: The scalability of GDPO to larger graphs (e.g., with 500 or more nodes) remains unexplored.

For broader impact, this paper presents work whose goal is to advance the field of Machine Learning. There are many potential societal consequences of our work, none which we feel must be specifically highlighted here.

## A.3 General Graph Generation

**Baselines.** There are several baseline methods for general graph generation, we summarize them as follows:

- GraphRNN: a deep autoregressive model designed to model and generate complex distributions over graphs. It addresses challenges like non-uniqueness and high dimensionality by decomposing the generation process into node and edge formations.

- SPECTRE: a novel GAN for graph generation, approaches the problem spectrally by generating dominant parts of the graph Laplacian spectrum and matching them to eigenvalues and eigenvectors. This method allows for modeling global and local graph structures directly, overcoming issues like expressivity and mode collapse.

- GDSS: A novel score-based generative model for graphs is introduced to tackle the task of capturing permutation invariance and intricate node-edge dependencies in graph data generation. This model employs a continuous-time framework incorporating a novel graph diffusion process,

**Algorithm 1:** Graph Diffusion Policy Optimization

**Input:** graph DPM $p_\theta$
**Input:** # of diffusion steps $T$, # of timestep samples $|\mathcal{T}|$
**Input:** reward signal $r(\cdot)$, # of trajectory samples $K$
**Input:** learning rate $\eta$ and # of training steps $N$
**Output:** Final graph DPM $p_\theta$
**for** $i = 1, \ldots, N$ **do**
  **for** $k = 1, \ldots, K$ **do**
    $\boldsymbol{G}_{0:T}^{(k)} \sim p_\theta$                                              `// Sample trajectory`
    $\mathcal{T}_k \sim \text{Uniform}(\llbracket 1, T \rrbracket)$                                    `// Sample timesteps`
    $r_k \leftarrow r(\boldsymbol{G}_0^{(k)})$                                                `// Get rewards`
  `// Estimate reward mean and variance`
  $\bar{r} \leftarrow \frac{1}{K} \sum_{k=1}^{K} r_k \qquad \text{std}[r] \leftarrow \sqrt{\frac{\sum_{k=1}^{K}(r_k - \bar{r})^2}{K-1}}$
  `// Estimate the eager policy gradient`
  $\boldsymbol{g}(\theta) \leftarrow \frac{1}{K} \sum\limits_{k=1}^{K} \frac{T}{|\mathcal{T}_k|} \sum\limits_{t \in \mathcal{T}_k} (\frac{r_k - \bar{r}}{\text{std}[r]}) \nabla_\theta \log p_\theta(\boldsymbol{G}_0^{(k)} | \boldsymbol{G}_t^{(k)})$
  `// Update model parameter`
  $\theta \leftarrow \theta + \eta \cdot \boldsymbol{g}(\theta)$

characterized by stochastic differential equations (SDEs), to simultaneously model distributions of nodes and edges.

- DiGress: DiGress is a discrete denoising diffusion model designed for generating graphs with categorical attributes for nodes and edges. It employs a discrete diffusion process to iteratively modify graphs with noise, guided by a graph transformer network. By preserving the distribution of node and edge types and incorporating graph-theoretic features, DiGress achieves state-of-the-art performance on various datasets.

- MOOD: MOOD introduces Molecular Out-Of-distribution Diffusion, which employs out-of-distribution control in the generative process without added costs. By incorporating gradients from a property predictor, MOOD guides the generation process towards molecules with desired properties, enabling the discovery of novel and valuable compounds surpassing existing methods.

**Metrics.** The metrics of general graph generations are all taken from GraphRNN [38]. The reported metrics compare the discrepancy between the distribution of certain metrics on a test set and the distribution of the same metrics on a generated graph. The metrics measured include degree distributions, clustering coefficients, and orbit counts (which measure the distribution of all substructures of size 4). Following DiGress [61], we do not report raw numbers but ratios computed as follows:

$$r = \text{MMD}(\textit{generated}, \textit{test})^2 / \text{MMD}(\textit{training}, \textit{test})^2 \tag{13}$$

Besides, we explain some metrics that are used in the general graph generation:

- *Clus*: the clustering coefficient measures the tendency of nodes to form clusters in a network. Real-world networks, especially social networks, often exhibit tightly knit groups with more ties between nodes than expected by chance. There are two versions of this measure: global, which assesses overall clustering in the network, and local, which evaluates the clustering around individual nodes.

- *Orb*: Graphlets are induced subgraph isomorphism classes in a graph, where occurrences are isomorphic or non-isomorphic. They differ from network motifs, which are over- or under-represented graphlets compared to a random graph null model. Orb will count the occurrences of each type of graphlet in a graph. Generally, if two graphs have similar numbers of graphlets, they are considered to be relatively similar.

### A.4 Molecule Property Optimization

**Implementation Details.** Following FREED [67], we selected five proteins, PARP-1 (Poly [ADP-ribose] polymerase-1), FA7 (Coagulation factor VII), 5-HT1B (5-hydroxytryptamine receptor 1B),

BRAF (Serine/threonine-protein kinase B-raf), and JAK2 (Tyrosine-protein kinase JAK2), which have the highest AUROC scores when the protein-ligand binding affinities for DUD-E ligands are approximated with AutoDock Vina [13], as the target proteins for which the docking scores are calculated. QED and SA scores are computed using the RDKit library.

**Baselines.** There are several baseline methods for molecular graph generation under the given objectives, they are diverse in methodology and performance, we summarize them as follows:

- GCPN: Graph Convolutional Policy Network (GCPN) is a general graph convolutional network-based model for goal-directed graph generation using reinforcement learning. The GCPN is trained to optimize domain-specific rewards and adversarial loss through policy gradient, operating within an environment that includes domain-specific rules.

- REINVENT: This method enhances a sequence-based generative model for molecular design by incorporating augmented episodic likelihood, enabling the generation of structures with specified properties. It successfully performs tasks such as generating analogs to a reference molecule and predicting compounds active against a specific biological target.

- HierVAE: a hierarchical graph encoder-decoder for drug discovery, overcoming limitations of previous approaches by using larger and more flexible graph motifs as building blocks. The encoder generates a multi-resolution representation of molecules, while the decoder adds motifs in a coarse-to-fine manner, effectively resolving attachments to the molecule.

- FREED: a novel reinforcement learning (RL) framework for generating effective acceptable molecules with high docking scores, crucial for drug design. FREED addresses challenges in generating realistic molecules and optimizing docking scores through a fragment-based generation method and error-prioritized experience replay (PER).

- MOOD: please refer to Appendix A.3.

**Metrics.** There are several metrics for evaluating the molecule properties, we summarize the meaning of these metrics as follows:

- Docking Score: Docking simulations aim to find the best binding mode based on scoring functions. Scoring functions in computational chemistry and molecular modeling predict binding affinity between molecules post-docking. They are commonly used for drug-protein interactions, but also for protein-protein or protein-DNA interactions. After defining the score function, we can optimize to find the optimal drug-protein matching positions and obtain the docking score.

- *QED*: Drug-likeness evaluation in drug discovery often lacks nuance, leading to potential issues with compound quality. We introduce QED, a measure based on desirability, which considers the distribution of molecular properties and allows the ranking of compounds by relative merit. QED is intuitive, transparent, and applicable to various settings. We extend its use to assess molecular target druggability and suggest it may reflect aesthetic considerations in medicinal chemistry.

- *SA*: a scoring method for rapid evaluation of synthetic accessibility, considering structural complexity, similarity to available starting materials, and strategic bond assessments. These components are combined using an additive scheme, with weights determined via linear regression analysis based on medicinal chemists' accessibility scores. The calculated synthetic accessibility values align well with chemists' assessments.

## A.5  Additional Results of the GDPO

Table 5: General graph generation on SBM and Planar datasets with different reward signals.

| METHOD | PLANAR GRAPHS | | | |
|---|---|---|---|---|
| | Deg ↓ | Clus ↓ | Orb ↓ | V.U.N (%) ↑ |
| *Validity* (0.6) | $0.03 \pm 0.03$ | $0.54 \pm 0.08$ | $0.02 \pm 0.01$ | $72.34 \pm 2.78$ |
| *Validity* (0.7) | $0.03 \pm 0.04$ | $0.62 \pm 0.11$ | $0.02 \pm 0.01$ | $73.83 \pm 2.49$ |
| *Validity* (0.8) | $0.12 \pm 0.04$ | $0.88 \pm 0.34$ | $0.24 \pm 0.07$ | $78.68 \pm 3.12$ |
| *Validity* (0.9) | $0.86 \pm 0.12$ | $2.17 \pm 0.84$ | $1.46 \pm 0.78$ | $81.26 \pm 3.02$ |

**Study of the Reward Signals.** In Table. 5, we showcase the performance of GDPO on Planar under different configurations of reward weights. We keep the three weights related to distance the same

and adjust the weight of validity while ensuring that the sum of weights is 1. The results indicate that GDPO is not very sensitive to the weights of several reward signals for general graph generation, even though these weight configurations vary significantly, they all achieve good performance. Additionally, we found that GDPO can easily increase *V.U.N* to above 80 while experiencing slight losses in the other three indicators. When applying GDPO in practice, one can make a tradeoff between them based on the specific application requirements.

Table 6: Study of the Important Sampling on ZINC250k.

| METHOD | METRIC | TARGET PROTEIN | | | | |
| --- | --- | --- | --- | --- | --- | --- |
| | | *parp1* | *fa7* | *5ht1b* | *braf* | *jak2* |
| DDPO | *Hit Ratio* | $0.419 \pm 0.280$ | $0.342 \pm 0.685$ | $5.488 \pm 1.989$ | $0.445 \pm 0.297$ | $1.717 \pm 0.684$ |
| | *DS (top 5%)* | $-9.247 \pm 0.242$ | $-7.739 \pm 0.244$ | $-9.488 \pm 0.287$ | $-9.470 \pm 0.373$ | $-8.990 \pm 0.221$ |
| DDPO-IS | *Hit Ratio* | $0.945 \pm 0.385$ | $0.319 \pm 0.237$ | $10.304 \pm 1.277$ | $0.436 \pm 0.272$ | $2.697 \pm 0.462$ |
| | *DS (top 5%)* | $-9.633 \pm 0.206$ | $-7.530 \pm 0.225$ | $-9.877 \pm 0.174$ | $-9.468 \pm 0.252$ | $-9.120 \pm 0.149$ |
| GDPO-IS | *Hit Ratio* | $0.850 \pm 0.602$ | $0.826 \pm 0.827$ | $16.283 \pm 1.190$ | $1.339 \pm 0.392$ | $4.381 \pm 0.501$ |
| | *DS (top 5%)* | $-9.482 \pm 0.300$ | $-8.254 \pm 0.180$ | $-10.361 \pm 0.319$ | $-9.771 \pm 0.120$ | $-9.583 \pm 0.202$ |
| GDPO (OURS) | *Hit Ratio* | $\mathbf{9.814} \pm 1.352$ | $\mathbf{3.449} \pm 0.188$ | $\mathbf{34.359} \pm 2.734$ | $\mathbf{9.039} \pm 1.473$ | $\mathbf{13.405} \pm 1.151$ |
| | *DS (top 5%)* | $-\mathbf{10.938} \pm 0.042$ | $-\mathbf{8.691} \pm 0.074$ | $-\mathbf{11.304} \pm 0.093$ | $-\mathbf{11.197} \pm 0.132$ | $-\mathbf{10.183} \pm 0.124$ |

**The Impact of Important Sampling.** The importance sampling technique in DDPO, aims to facilitate multiple steps of optimization using the same batch of trajectories. This is achieved by weighting each item on the trajectory with an importance weight derived from the density ratio estimated using the model parameters from the previous step $\theta_{\text{prev}}$ and the current step $\theta$ (referred to as DDPO-IS):

$$\nabla_\theta \mathcal{J}_{\text{DDPO-IS}}(\theta) = E_\tau \left[ r(G_0) \sum_{t=1}^T \frac{p_\theta(G_{t-1}|G_t)}{p_{\theta_{\text{prev}}}(G_{t-1}|G_t)} \nabla_\theta \log p_\theta(G_{t-1}|G_t) \right]. \tag{14}$$

Our eager policy gradient, independently motivated, aims to address the high variance issue of the policy gradient in each step of optimization, as elaborated in Sec. 4.2. Intuitively, the eager policy gradient can be viewed as a biased yet significantly less fluctuating gradient estimation.

We conducted a series of experiments on ZINC250k to compare DDPO, DDPO-IS, and GDPO. The experimental setup remains consistent with the description in Section 6.2. Additionally, considering that the importance sampling technique in DDPO and our eager policy gradient appear to be orthogonal, we also explored combining them simultaneously (referred to as GDPO-IS):

$$\nabla_\theta \mathcal{J}_{\text{GDPO-IS}}(\theta) = E_\tau \left[ r(G_0) \sum_{t=1}^T \frac{p_\theta(G_0|G_t)}{p_{\theta_{\text{prev}}}(G_0|G_t)} \nabla_\theta \log p_\theta(G_0|G_t) \right]. \tag{15}$$

In Table. 6, while importance sampling enhances the performance of DDPO, consistent with the results reported in the DDPO paper, it does not yield improvements for GDPO-IS over GDPO. We speculate that this discrepancy may be due to the biasness of the eager policy gradient, rendering it incompatible with the importance sampling technique. We intend to investigate the mechanism and address this in our future work. Nevertheless, it is noteworthy that the performance of DDPO-IS remains inferior to GDPO, indicating the superiority of our proposed GDPO method.

Table 7: Novelty and Diversity on ZINC250k.

| METRIC | TARGET PROTEIN | | | | |
| --- | --- | --- | --- | --- | --- |
| | *parp1* | *fa7* | *5ht1b* | *braf* | *jak2* |
| IOU | 0.0763% | 0.0752% | 0.0744% | 0.113% | 0.0759% |
| UNIQ | 94.86% | 97.35% | 99.86% | 99.74% | 97.02% |

**Novelty and Diversity of GDPO.** To provide further insight into the novelty and diversity of our approach, we introduce two additional metrics:

- Intersection over Union (IoU): We compare two sets of molecules: 1) 500 molecules generated by GDPO (denoted as GDPO) and 2) top 500 molecules among 10,000 molecules generated by our base DPM before finetuning (denoted as TopPrior). We then compute IoU=$100 \times \frac{|\text{GDPO} \cap \text{TopPrior}|}{|\text{GDPO} \cup \text{TopPrior}|}$%. We report an average IoU of 5 independent runs.

- Uniqueness in 10k samples (Uniq): We generate 10,000 molecules and compute the ratio of unique molecules $\text{Uniq} = 100 \times \frac{\text{\# unique molecules}}{\text{\# all molecules}}\%$.

In Table. 7, these results show that GDPO has not converged to a trivial solution, wherein it merely selects a subset of molecules generated by the prior diffusion model. Instead, GDPO has learned an effective and distinct denoising strategy from the prior diffusion model.

**The Gap between Image DPMs and Graph DPMs.** GDPO is tackling the high variance issue inherent in utilizing policy gradients on graph DPMs, as stated and discussed in Sec. 4.2. To provide clarity on what GDPO tackles, we would like to elaborate more on the high variance issue of policy gradients on graph DPMs. Consider the generation trajectories in image and graph DPMs:

In image DPMs, the generation process follows a (discretization of) continuous diffusion process $(\mathbf{x}_t)_{t\in[0,T]}$. The consecutive steps $\mathbf{x}_{t-1}$ and $\mathbf{x}_t$ are typically close due to the Gaussian reverse denoising distribution $p(\mathbf{x}_{t-1}|\mathbf{x}_t)$ (typically with a small variance).

In graph DPMs, the generation process follows a discrete diffusion process $(G_T, \ldots, G_0)$, where each $G_t$ is a concrete sample (i.e., one-hot vectors) from categorical distributions. Therefore, consecutive steps $G_{t-1}$ and $G_t$ can be very distant. This makes the trajectory of graph DPMs more fluctuating than images and thus leads to a high variance of the gradient $\nabla_\theta \log p(G_{t-1}|G_t)$ (and the ineffectiveness of DDPO) when evaluated with same number of trajectories as in DDPO.

Regarding the "distance" between two consecutive steps $G_t$ and $G_{t-1}$, our intuition stems from the fact that graphs generation trajectories are inherently discontinuous. This means that each two consecutive steps can differ significantly, such as in the type/existence of edges. In contrast, the generation trajectories of images, governed by reverse SDEs, are continuous. This continuity implies that for fine-grained discretization (i.e., large $T$), $\mathbf{x}_t$ and $\mathbf{x}_{t-1}$ can be arbitrarily close to each other (in the limit case of $T \to \infty$).

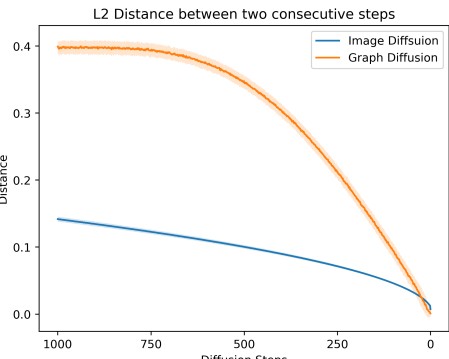

Figure 4: We investigate the L2 distance between two consecutive steps in two types of DPMs. The diffusion step is 1000 for two models.

To provide quantitative support for this discussion, we conduct an analysis comparing the distances between consecutive steps in both image and graph DPMs. We employ a DDPM [a] pre-trained on CIFAR-10 for image diffusion and DiGress [b] pre-trained on the Planar dataset for graph diffusion, both with a total of $T = 1000$ time steps. In these models, graphs are represented with one-hot vectors (as described in Sec. 3) and image pixels are rescaled to the range $[0, 1]$, ensuring their scales are comparable. We then directly compare the per-dimension L2 distances in both spaces, denoted as $\|G_t - G_{t-1}\|_2 / \sqrt{D_G}$ and $\|\mathbf{x}_t - \mathbf{x}_{t-1}\|_2 / \sqrt{D_I}$, where $D_G$ and $D_I$ are the dimensions of graphs and images, respectively. (Dividing by $\sqrt{D}$ is to eliminate the influence of different dimensionalities.) We sample 512 trajectories from each DPM and plot the mean and deviation of distances with respect to the time step $t$.

In Fig. 4, the results support the explanation of GDPO. While we acknowledge that graphs and images reside in different spaces and typically have different representations, we believe the comparison with L2 distance can provide valuable insights into the differences between graph and image DPMs.

**GDPO on the Synthetic Tree-like Dataset.** We first generate a tree and then connect a clique to the nodes of the tree, performing a specified number of rewrite operations as suggested. Based on

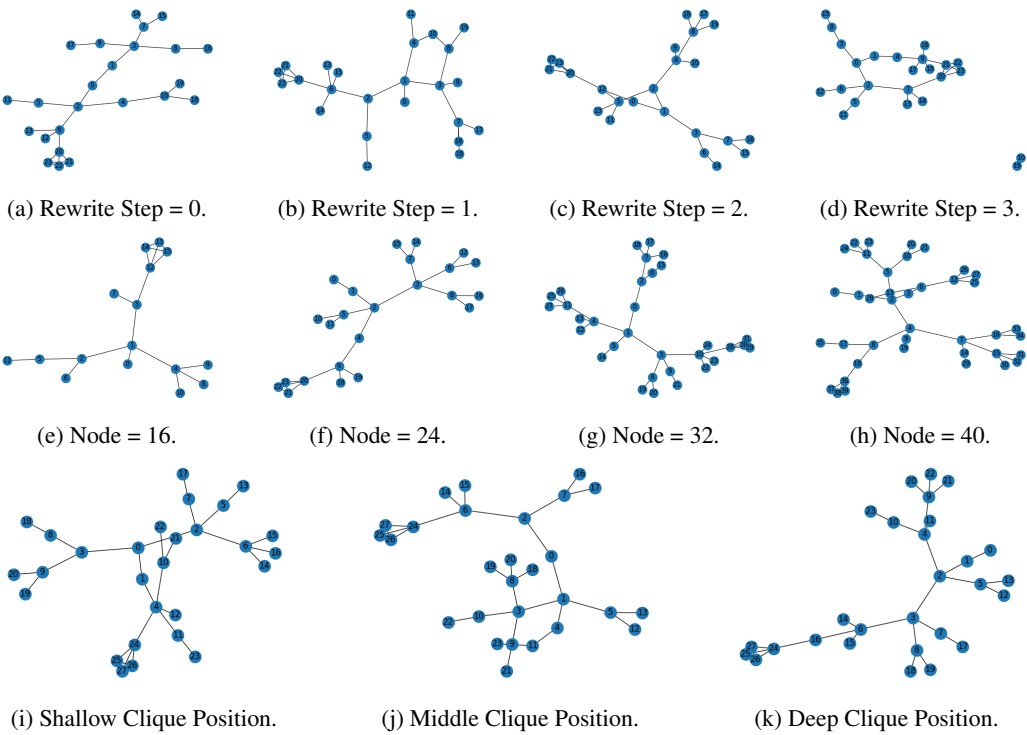

(a) Rewrite Step = 0.  (b) Rewrite Step = 1.  (c) Rewrite Step = 2.  (d) Rewrite Step = 3.

(e) Node = 16.  (f) Node = 24.  (g) Node = 32.  (h) Node = 40.

(i) Shallow Clique Position.  (j) Middle Clique Position.  (k) Deep Clique Position.

Figure 5: Tree with Different Parameters. Node 0 is the root node.

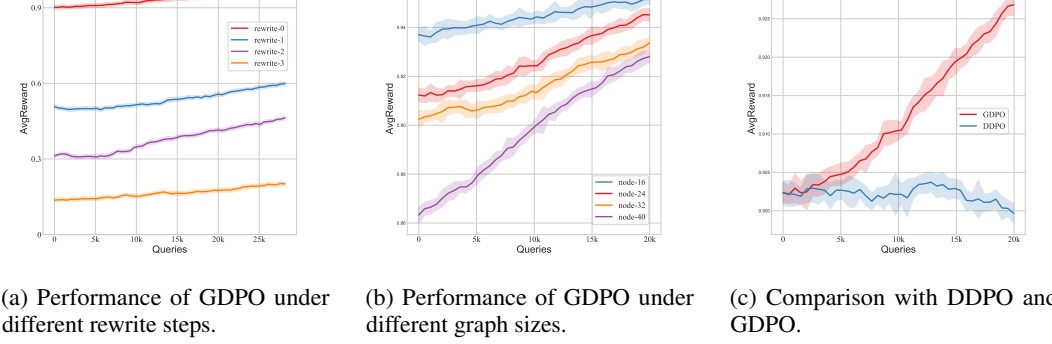

(a) Performance of GDPO under different rewrite steps.

(b) Performance of GDPO under different graph sizes.

(c) Comparison with DDPO and GDPO.

Figure 6: Ablation Study on the Synthetic Tree-like Dataset.

the number of rewrite steps, graph size, and clique position, we generate multiple datasets, each containing 400 samples. Of these, 256 samples are used for training Graph DPMs, with the remaining samples allocated for validation and testing. In Fig. 5, we present some examples. Fig. 5(a)illustrates a tree structure with a clique of size 4. When the number of rewrite steps is 3, Fig. 5(d) demonstrates that the overall structure of the samples is disrupted. After training the Graph DPMs, we apply GDPO. The model receives a reward of 1 when it generates a tree with a clique; otherwise, the reward is 0. We then ablate the following factors to test the performance of GDPO.

Rewrite Steps: In Fig. 6(a), we demonstrate GDPO's performance across different rewrite steps, with four curves representing steps ranging from 0 to 3. Despite a notable decrease in the initial reward as the number of rewrite steps increases, GDPO consistently optimizes the Graph DPMs effectively to generate the desired graph structure.

Graph Size: In Fig. 6(b), we gradually increase the number of nodes from 16 to 40. The results show that graph size affects the initial reward but does not impact GDPO's optimization performance.

Clique Position: We experiment with inserting the clique at different levels of the tree but find no significant difference. We believe this is because the position of the clique does not affect the initial reward of the Graph DPMs, leading to similar optimization results with GDPO.

Comparison with Baseline: In Fig. 6(c), we compare GDPO with DDPO. The results, consistent with those in Figure 2 of the paper, reveal a clear distinction between GDPO and DDPO in handling challenging data generation tasks.

### A.6 Discussions

**Comparison with the $x_0$-prediction Formulation.** Indeed, our eager policy gradient in Eq. 10, compared to the policy gradient of REINFORCE in Eq. 8, resembles the idea of training a denoising network to predict the original uncorrupted graph rather than performing one-step denoising. However, we note that training a denoising network to predict the original data is fundamentally a matter of parametrization of one-step denoising. Specifically, the one-step denoising $p_\theta(x_{t-1}|G_t)$ is parameterized as a weighted sum of $x_0$-prediction, as described in Eq. 1. Our method in Eq. 8 is motivated differently, focusing on addressing the variance issue as detailed in Sections 4.2 and 4.3.

**Pros and Cons of the RL Approach against Classifier-based and Classifier-free Guidance for Graph DPMs.** Compared to graph diffusion models using classifier-based and classifier-free guidance, RL approaches such as GDPO have at least two main advantages:

- Compatibility with discrete reward signals and discrete graph representations: As guidance for diffusion models is based on gradients, a differentiable surrogate (e.g., property predictors [65, 37]) is needed for non-differentiable reward signals (e.g., results from physical simulations). RL approaches naturally accommodate arbitrary reward functions without the need for intermediate approximations.
- Better sample efficiency: For graph diffusion models with classifier-based or classifier-free guidance, labeled data are required at the beginning and are independently collected with the graph diffusion models. In contrast, RL approaches like GDPO collect labeled data during model training, thus allowing data collection from the current model distribution, which can be more beneficial. We also empirically observe a significant gap in sample efficiency.

**Analysis on the Bias-variance Trade off.** The main bias of GDPO arises from modifying the "weight" term in Eq. 9, which shifts the model's focus more towards the generated results rather than the intermediate process, thereby reducing potential noise. Due to the discrete nature of Graph DPMs, the $x_0$-prediction and $x_{t-1}$-prediction formulations cannot be related through denoising objectives as in continuous DPMs. This issue also complicates the connection between DDPO and GDPO. We have not yet identified a relevant solution and are still working on it. In our empirical study, we do not observe significant performance variance and tradeoff for GDPO given the current scale of experiments. This may be due to the graph sizes we explored not being sufficiently large. In future implementations, we will incorporate support for sparse graphs to assess GDPO's performance on larger graph datasets and investigate the tradeoff more thoroughly.

