# OpenReview forum: "Graph Diffusion Policy Optimization"
_NeurIPS.cc/2024/Conference — NeurIPS 2024 poster_

### Official Review · Reviewer_4R3n · 2024-07-07

**Soundness:** 3
**Presentation:** 3
**Contribution:** 3
**Rating:** 6
**Confidence:** 4

**Summary:**

The paper studies the problem of learning graph diffusion generative models on arbitrary non-differentiable objectives using policy gradients. Authors argue that the recently proposed DDPO technique doesn't work well on the discrete, graph-related learning tasks and consider a modified objective and a corresponding gradient estimate which they refer to as Graph Diffusion Policy Optimization (GDPO). They show that GDPO performs significantly better on a number of reward functions and datasets.

**Strengths:**

## GDPO seems to be efficient

Discrete diffusion models are valuable class of machine learning models and being able to trained them on non-differentiable objectives might might unlock interesting applications, in particular, in optimizing molecular graphs as authors demonstrated.

## The paper is well-written and easy to follow

I can clearly answer most of my own questions about how GDPO works and it should be easy to reproduce main results.

**Weaknesses:**

## Underexplored potential of modern RL techniques

While the proposed GDPO is valueable regardless, it is unclear to me that the biased objective / gradient estimate of eager gradients is necessary. Neither DDPO nor this paper explores even the simplest techniques such as actor-critic, PPO / TRPO or more sophisticated versions of importance sampling. DDPO showed good performance on image tasks without such advances and it could well be that it would still work well on graph tasks.

## Lack of bias-variance analysis

Authors acknowledge that eager gradients is a biased version of the standard policy gradients but don't provider either theoretical or empirical analysis of the bias-variance tradeoff between the two methods.

In the appendix (line 660) authors a strange statement that importance sampling is used to reduce variance. It is not obvious to me all. Importance sampling can both reduce or increase variance depending on the proposal or the policy generating the trajectory. What it achieves is that it allows to train on experiences generated by policies other than the current policy being optimized.

**Questions:**

My main question is what is the nature of the bias the GDPO introduces? Is there a reasonable objective that is followed go eager gradients? Can we still interpret them as policy gradients in some kind of modified MDP?

Figure 4: what is $D_G$ and $D_I$ concretely? Does it make sense to assess $L_2$ distances on discrete objects with categorical features?

What happens if you apply GDPO to image diffusion? Does it still work or work better?

**Limitations:**

See above, I believe the paper needs a clearer discussion of the bias introduced by policy gradients. At the moment, it is not clear what is the connection of GDPO to the reverse diffusion MDP.

---

> ### Author Rebuttal · Authors · 2024-08-07
>
> Thank you for your insightful review and valuable questions. Below, we respond to the comments in ***Weaknesses (W)*** and ***Questions (Q)***.
>
> ***W1: Other RL techniques***
>
>
> Thank you for your suggestions. Due to the multi-step generation process characteristics of DPMs, directly estimating model gradients from rewards is very challenging. While we acknowledge the theoretical limitations of GDPO compared to DDPO, our results suggest that GDPO is an effective method for discrete DPMs and fills a gap in related research. Exploring other RL techniques, such as actor-critic, PPO, TRPO, or more sophisticated versions of importance sampling, to further enhance GDPO should be an interesting extension, which we leave to future work.
>
> ---
>
> ***W2&Q1: Analysis on bias***
>
> Thank you for highlighting these limitations. The main bias of GDPO arises from modifying the "weight" term in Eq. (9), which shifts the model's focus more towards the generated results rather than the intermediate process, thereby reducing potential noise. Due to the discrete nature of Graph DPMs, the $x\_0$-prediction and $x\_\{t-1\}$-prediction formulations cannot be related through denoising objectives as in continuous DPMs. This issue also complicates the connection between DDPO and GDPO. We have not yet identified a relevant solution and are still working on it.
>
> In our empirical study, we do not observe significant performance variance and tradeoff for GDPO given the current scale of experiments. This may be due to the graph sizes we explored not being sufficiently large. In future implementations, we will incorporate support for sparse graphs to assess GDPO's performance on larger graph datasets and investigate the tradeoff more thoroughly.
>
> Regarding the statement about importance sampling, we apologize for any misunderstanding. In line 660, we did not claim that importance sampling techniques can reduce variance. Instead, we stated that they can be used to update DPMs multiple times with the same batch of trajectories, which aligns with your understanding, i.e., training on experiences generated by policies other than the current policy being optimized.
>
> We will include the above discussion in the final version.
>
> ---
>
> ***Q2: Clarification on Figure 4***
>
> We apologize for the lack of clarity. $D_I$ and $D_G$ represent the feature dimensions of images and graphs, respectively. For example, if an image has a size of $3 \times 32 \times 32$, then $D_I = 3072$. For a graph, $D_G$ is the product of the number of nodes and the feature dimension of the nodes. Since the L2 norm sums over the feature dimensions, we average over these dimensions to eliminate the influence of dimensionality. We choose the L2 norm to maintain consistency in the metric for comparisons, we acknowledge that graphs and images reside in different spaces and typically have different representations, we believe the comparison with L2 distance can provide valuable insights into the differences between graph and image DPMs.
>
> ---
>
> ***Q3: GDPO on images***
>
> Thank you for your insightful question. Our research primarily focuses on graph DPMs, so we did not include experiments on Image DPMs. We attempted to adapt GDPO to Image DPMs but observed no significant advantages. We believe that for continuous DPMs, the policy gradient noise of DDPO is already sufficiently small, and there is no need to adjust the weight term. We will include this discussion in the final version.

---

> > ### Comment · Reviewer_4R3n · 2024-08-12
> >
> > I thank authors for their response which I found helpful but ultimately not convincing enough for me to raise my score.

---

> > > ### Author Response · Authors · 2024-08-12
> > >
> > > We appreciate your valuable feedback. We will further polish the paper and incorporate the rebuttal discussions into the final revision. Thank you!

---

### Official Review · Reviewer_hRfp · 2024-07-12

**Soundness:** 3
**Presentation:** 3
**Contribution:** 3
**Rating:** 6
**Confidence:** 4

**Summary:**

This paper introduces graph diffusion policy optimization (GDPO), a policy gradient method for optimizing graph diffusion probabilistic models with respect to non-differentiable reward signals. By establishing the connection between a T-step denoising process and a T-step Markov Decision Process (MDP), policy gradient methods can be applied to graph diffusion models. While a previous work (DDPO) report s competitive generation quality for image diffusion models trained with the classical REINFORCE algorithm, the authors empirically observe a convergence issue when applying REINFORCE to graph diffusion models. This issue is possibly due to the increasingly vast space constituted by discrete graph trajectories as the number of nodes in the graph increases. To address this issue, the authors propose GDPO, a modified policy optimization objective. Empirical studies demonstrate the effectiveness of the proposed modified objective.

**Strengths:**

**S1.** The proposed methodology is neat and well-motivated.

**S2.** The paper is well-written and easy to follow.

**S3.** The empirical studies demonstrate the effectiveness of the proposed approach.

**Weaknesses:**

**W1.** Some experiment settings are not completely clear from the description. See the questions below.

**W2.** The proposed modification coincides with the idea of training a denoising network to predict the original uncorrupted graph rather than perform one-step denoising. Some discussions are expected.

**W3.** Classifier-based and classifier-free guidance are two popular approaches for training conditional diffusion models and have been previously explored for graph diffusion models. Some discussions on the potential pros and cons of the RL approach against them are expected.

**Questions:**

**Q1.** For the graph diffusion baselines considered, are they trained using classifier-free or classifier-based guidance?

**Q2.** A key question in understanding the pros and cons of the different conditional generation approaches is sample efficiency. If ground truth reward functions are employed for GDPO training, then similarly we can label the samples generated by non-RL conditional graph diffusion models as extra training samples. The key question is then how the performance of the different models changes against the number of reward function evaluations.

**Limitations:**

The paper considers discrete time graph diffusion models. Whether the proposed approach is effective for continuous time graph diffusion models remains unexplored.

---

> ### Author Rebuttal · Authors · 2024-08-07
>
> Thank you for your insightful review and valuable suggestions. Below, we respond to the comments in ***Weaknesses (W)*** and ***Questions (Q)***.
>
>
> ---
>
> ***W1: Experimental settings***
>
> Thanks for your suggestions. We will continue to polish our introduction to experimental settings, moving important details from Appendix to the Sec. 6 and providing additional experimental details for clarify.
>
> ---
>
> ***W2: Comparison with the $x\_0$-prediction formulation***
>
> Thank you for pointing this out. Indeed, our eager policy gradient in Eq. (10), compared to the policy gradient of REINFORCE in Eq. (8), resembles the idea of training a denoising network to predict the original uncorrupted graph rather than performing one-step denoising. However, we note that training a denoising network to predict the original data is fundamentally a matter of *parametrization* of one-step denoising. Specifically, the one-step denoising $p\_\\theta(x\_\{t-1\}|G\_t)$ is parameterized as a weighted sum of $x\_0$-predictions $p\_\\theta(x\_\{0\}|G\_t)$, as described in Eq. (1). The eager policy gradient in Eq. (10) is motivated differently, focusing on *addressing the variance issue* as detailed in Sections 4.2 and 4.3. We will include this discussion in the final version.
>
> ---
>
> ***W3: Discussion on the potential pros and cons of the RL approach against classifier-based and classifier-free guidance for graph diffusion models***
>
>
> Compared to graph diffusion models using classifier-based and classifier-free guidance, RL approaches such as GDPO have at least two main advantages:
> - Compatibility with discrete reward signals and discrete graph representations (L24-30): As guidance for diffusion models is based on gradients, a differentiable surrogate (e.g., property predictors [29, 37]) is needed for non-differentiable reward signals (e.g., results from physical simulations). RL approaches naturally accommodate arbitrary reward functions without the need for intermediate approximations.
> - Better sample efficiency: For graph diffusion models with classifier-based or classifier-free guidance, labeled data are required at the beginning and are independently collected with the graph diffusion models. In contrast, RL approaches like GDPO collect labeled data during model training, thus allowing data collection from the current model distribution, which can be more beneficial. We also empirically observe a significant gap in sample efficiency. Please also see our response to ***Q2***.
>
> Potential cons of GDPO are discussed in the Limitations, see Appendix A.2. We will include the above discussion in the final version.
>
> ---
>
> ***Q1: Are the graph diffusion baselines trained using classifier-free or classifier-based guidance?***
>
> The graph diffusion baselines in our experiments, i.e., MOOD and DiGress-guidance, are both classifier-based methods. For these methods, additional regressors for guidance (referred to as property predictors) are trained on graph samples with ground truth rewards. Specifically, noise is added to the input molecular graph so that the predictors can provide correct guidance at all timesteps during the denoising process.
>
> For graph diffusion models using classifier-free guidance, we are currently not aware of comparable work and will be happy to include this in future work.
>
>
> ---
>
> ***Q2: Sample efficiency and performance comparison***
>
> In our experiments on ZINC250k, we used 100,000 extra samples with ground truth rewards to train property predictors to provide gradient guidance for the MOOD and DiGress-guidance baselines. Note that for GDPO, the total query of reward is only 20,000, which is much smaller than that used for guidance. Nonetheless, a significant performance gap remains between these methods and GDPO. This demonstrates that GDPO has much better sample efficiency compared to graph diffusion models based on guidance. We will highlight these results in the revision.

---

> > ### Comment · Reviewer_hRfp · 2024-08-07
> >
> > Thank you for your response, which has addressed most of my questions. I've increased the "Soundness score" from 2 to 3 and the rating from 5 to 6.

---

> > > ### Author Response · Authors · 2024-08-08
> > > **Thank you for your support and raising the score**
> > >
> > > We greatly appreciate your valuable feedback and the score improvement. We will further polish the paper and incorporate the rebuttal discussions into the final revision. Thank you!

---

### Official Review · Reviewer_gTVc · 2024-07-14

**Soundness:** 3
**Presentation:** 3
**Contribution:** 3
**Rating:** 7
**Confidence:** 4

**Summary:**

This paper introduces Graph Diffusion Policy Optimization (GDPO), a novel approach to optimize graph diffusion models for arbitrary objectives using reinforcement learning. The key contributions are:

1. Formulating the denoising process of graph diffusion probabilistic models (DPMs) as a Markov decision process and proposing an "eager policy gradient" method tailored for graph DPMs to address high variance issues in standard policy gradient approaches.
2. Demonstrating state-of-the-art performance on various graph generation tasks, including general graph generation and molecular graph generation with complex objectives.

The authors show that GDPO significantly outperforms baseline methods, including other graph generation techniques and adaptations of diffusion model optimization approaches from the image domain (e.g., DDPO).

**Strengths:**

1. Novelty and Originality:  Introduction of the "eager policy gradient" method, to address the high variance issues encountered with standard policy gradients in graph diffusion models. Despite the lack of theory supporting it, it's a clever  solution to a significant challenge in optimizing these models for arbitrary objectives.

2. Clarity:
   - Clear problem formulation: The authors provide a well-structured explanation of the challenges in optimizing graph DPMs and why existing methods fall short.
   - Effective visualization: Figure 1 offers a clear overview of the GDPO method, aiding understanding of the approach.
   - Detailed ablation studies: The paper includes thorough analyses of different components and configurations of GDPO, which helps clarify the contribution of each aspect of the method.

3. Significance:
   - Strong performance improvements: GDPO demonstrates substantial gains over state-of-the-art baselines across various graph generation tasks. For example, in molecular graph generation, it achieves up to a 19.31% improvement in hit ratio for generating effective drugs.
   - Sample efficiency: The method achieves good results with relatively few queries (e.g., 1/25 of the training samples), which is crucial for applications where reward evaluation may be computationally expensive, such as drug discovery.
   - Broad applicability: GDPO is flexible and can be applied to a wide range of graph generation tasks with complex, multi-objective reward functions. This versatility enhances its potential impact on the field.

4. Technical Quality:
   - Thorough experimentation: The authors provide extensive experiments on both general graph generation and molecular graph generation tasks, lending credibility to their claims.
   - Careful analysis of baseline methods: The paper includes a detailed study of  DDPO (a related method for image diffusion models) failing on graph DPMs, which strengthens the justification for GDPO.
   - Consideration of practical aspects: The authors address important practical considerations such as the impact of different reward weightings and the number of trajectories used for gradient estimation.

**Weaknesses:**

- Limited theoretical analysis: While the eager policy gradient is empirically effective, the paper lacks a rigorous theoretical treatment of its properties, particularly regarding the bias-variance trade-off.
-  The paper would benefit from a comparison to other RL-utilizing graph generation methods, particularly MolGAN https://arxiv.org/pdf/1805.11973 , which also applies RL techniques to molecular graph generation.
- Scalability concerns: The paper does not explore the method's performance on very large graphs (e.g., 500+ nodes), leaving questions about its scalability unanswered.
- Limited exploration of failure cases: While the authors provide a failure case related to novelty optimization, a more comprehensive exploration of scenarios where GDPO struggles would provide valuable insights into its limitations.

**Questions:**

1. How sensitive is GDPO to the choice of reward function? Are there certain types of rewards or objectives that are particularly challenging for the method (e.g., very sparse, high dynamic range, noisy...)?
2. I'd like to suggest an important ablation study on highly ambiguous graphs to better understand the limitations and robustness of GDPO. Specifically, I propose constructing a "noisy-tree-with-planted-motifs" dataset with the following process:

---

1. Generate a base tree structure with parameters:
   - Tree depth
   - Minimum fan-out factor
   - Maximum fan-out factor
   - Binned beta variation across the fan-out for a given node

2. Plant a tree of cliques as an expansion of  nodes at a chosen height in the base tree. This planted structure should have parameters:
   - Number of rings of cliques
   - Ring size and clique count

3. Apply a noisy rewiring process towards an Erdős-Rényi random graph with the same number of nodes and edges as the base graph. This process should take as parameters:
   - Number of rewiring steps
   - starting layer of the subgraph to be noised (starting at a given layer of the tree)

The noisy rewiring process would involve:
a) Randomly selecting an edge
b) Checking whether to remove it based on the probability of an edge existing in the target Erdős-Rényi graph
c) Selecting a random node pair to potentially add an edge
d) Repeating this process for the specified number of steps

I suggest ablating over:
1. The number of rewiring steps
2. The size of the subgraph to be noised
3. The overall graph size
4. The height at which the tree of cliques is planted
5. (optional) clique size and ring size

My hypothesis is that GDPO would struggle to maintain the planted tree of cliques structure and the overall tree structure as the noise level and graph size increase, as it would slowly fill up the capacity of the model with combinatorial structures. This ablation study would provide valuable insights into GDPO's performance on more complex and ambiguous graph structures.

Questions I'd like answered through this ablation study:

1. How does GDPO's performance change as the noise level (number of rewiring steps) increases?
2. What is the impact of graph size on GDPO's ability to maintain the planted tree of cliques and the overall tree structure?
3. Is there a critical point in the noise level or graph size where GDPO's performance significantly degrades?
4. How does the height at which the tree of cliques is planted affect GDPO's ability to maintain this structure? (i.e., how well does it maintain long-range dependeny)
5. How does GDPO compare to baseline methods on these more challenging graph structures?
6. Can you identify specific types of motifs or structures within the planted tree of cliques that GDPO struggles to maintain or generate in these noisy environments?

**Limitations:**

I think the limitations in the appendix are acceptable

---

> ### Author Rebuttal · Authors · 2024-08-07
>
> Thank you for your insightful comments and valuable suggestions. Below, we respond to the concerns raised in ***Weaknesses (W)*** and ***Questions (Q)***.
>
> ---
>
> ***W1: Theoretical Analysis, Scalability, and Failure Cases***
>
> Thank you for highlighting these points. Despite considerable efforts, theoretical analysis of the eager policy gradient remains challenging due to the discrete nature of Graph DPMs. We will continue to address this in future work. While GDPO shows significant efficiency, its scalability to larger graphs is constrained by the inherent limitations of graph DPMs, particularly concerning GPU memory. In future implementations, we aim to introduce support for sparse graphs to enhance scalability. Regarding failure cases, in addition to the case mentioned in Section 6.3, we believe that potential failure cases likely arise from tasks with very sparse reward signals. We will add further discussion to the final paper.
>
> ---
>
> ***W2: Comparison with RL-based graph generation methods***
>
> Thank you for your suggestion. We discussed MolGAN in the Introduction (line 32). We have also compared GDPO with several representative and leading RL-based Graph Generation Methods, such as GCPN, REINVENT, and FREED. These methods optimize graph neural networks to generate molecules with specific properties by designing molecule encoding schemes and utilizing modern RL techniques.
>
> ---
>
> ***Q1: Discussion on reward selection***
>
> In Appendix A.5,  Table 5, we investigated the sensitivity of GDPO to different rewards. The results demonstrate that adjusting the weights of sub-rewards does not significantly impact model performance, indicating that GDPO has a certain degree of robustness to reward variation. However, very sparse rewards generally pose a challenge. As detailed in Algorithm 1 in Appendix, GDPO normalizes the received rewards. For very sparse rewards (e.g., most rewards are zero and a few are one), the normalized values can become very large, leading to instability in optimization. To mitigate this issue, we follow DDPO in clipping gradients and discarding gradients generated by extremely large rewards. However, for extreme cases, these techniques may also fail. We will include further discussion on this topic in the final version.
>
> ---
>
> ***Q2: Ablation study on noisy tree with planted motifs***
>
> Thank you for sharing the idea. Based on your proposed setting, we conduct additional experiments. Due to time constraints, we do not perform extensive parameter exploration. Additionally, we made some simplifications to facilitate implementation and better explore the key factors. We first generate a tree and then connect a clique to the nodes of the tree, performing a specified number of rewrite operations as suggested. Based on the number of rewrite steps, graph size, and clique position, we generate multiple datasets, each containing 400 samples. Of these, 256 samples are used for training Graph DPMs, with the remaining samples allocated for validation and testing.
>
> In $\\textrm{\\color{blue}Figure A}$ of the rebuttal PDF, we present some examples. $\\textrm{\\color{blue} Figure A(a)}$ illustrates a tree structure with a clique of size 4. When the number of rewrite steps is 3, $\\textrm{\\color{blue} Figure A(d)}$ demonstrates that the overall structure of the samples is disrupted.
>
> After training the Graph DPMs, we apply GDPO. The model receives a reward of 1 when it generates a tree with a clique; otherwise, the reward is 0.
>
> **Rewrite Steps** In $\\textrm{\\color{blue} Figure B(a)}$, we demonstrate GDPO's performance across different rewrite steps, with four curves representing steps ranging from 0 to 3. Despite a notable decrease in the initial reward as the number of rewrite steps increases, GDPO consistently optimizes the Graph DPMs effectively to generate the desired graph structure.
>
> **Graph Size** In $\\textrm{\\color{blue} Figure B(b)}$, we gradually increase the number of nodes from 16 to 40. The results show that graph size affects the initial reward but does not impact GDPO’s optimization performance.
>
> **Critical Point** In the current Ablation Study, we do not observe any clear change points. This may be due to the limited range of parameters explored. We will include additional experiments in the final version to address this.
>
> **Clique Position** We experiment with inserting the clique at different levels of the tree but find no significant difference. We believe this is because the position of the clique does not affect the initial reward of the Graph DPMs, leading to similar optimization results with GDPO.
>
> **Comparison with Baseline** In $\\textrm{\\color{blue} Figure B(c)}$, we compare GDPO with DDPO. The results, consistent with those in  Figure 2 of the paper, reveal a clear distinction between GDPO and DDPO in handling challenging data generation tasks.
>
> **Challenging Motifs** Based on the current results, we do not observe related behaviors. However, as indicated by the results in  Table 1 of the paper, generating planar graphs with specific macro-topological features is quite challenging for GDPO. This is due to the model’s need to perform global summarization of the graph structure, and the rewards are typically sparse.
>
> Thank you very much for your suggestions. We will include the above discussions in the final version of the paper.

---

> > ### Comment · Reviewer_gTVc · 2024-08-11
> >
> > Thank your for performing the ablation study and addressing my other points. I am surprised and intrigued by the results of the ablation study and am now curious on how it would look like when investing a larger compute amount to deeper/wider graphs, but I fully understand this is not feasible in this round of investigation.  Since I already had the highest score across reviewers,  I will retain my score as is for now, but I  think this and the other additions have  made the paper stronger and I'm now quite comfortable with my score.

---

> > > ### Author Response · Authors · 2024-08-11
> > > **Thank you for your feedback**
> > >
> > > Thank you for your timely response and for acknowledging our efforts. We appreciate your positive feedback, especially regarding the strengthened contribution of the ablation study. We will further explore deeper/wider graphs and incorporate these findings, along with the rebuttal discussions, into the final revision. Thank you once again!

---

### Author Rebuttal · Authors · 2024-08-07

Dear Reviewers,

  We thank all reviewers for their constructive feedback, and we have responded to each reviewer individually. We have also uploaded a **Rebuttal PDF** that includes:

  $\\textrm{\\color{blue}Figure A}$: Some visualizations of graph data generated based on Reviewer gTVc's suggestions.

  $\\textrm{\\color{blue}Figure B}$: Results of the ablation study on the synthetic data.

---

### Decision · Program_Chairs · 2024-09-25

**Decision:**

Accept (poster)

**Comment:**

The paper proposes a methodology to optimize graph diffusion probabilistic models for arbitrary reward signals. The methodology has several novel aspects including the Markov Decision Process formulation and the policy gradient considered. Despite the fact that the paper lacks strong theoretical support, the empirical results are quite strong.  The response of the authors to the different questions raised further clarified several aspects of the paper. Therefore, I recommend accepting the paper.